# *Saccharomyces cerevisiae* DJ-1 paralogs maintain genome integrity through glycation repair of nucleic acids and proteins

**Gautam Susarla, Priyanka Kataria[†], Amrita Kundu[†], Patrick D'Silva\***

Department of Biochemistry, Indian Institute of Science, Bangalore, India

**Abstract** Reactive carbonyl species (RCS) such as methylglyoxal and glyoxal are potent glyco-lytic intermediates that extensively damage cellular biomolecules leading to genetic aberration and protein misfolding. Hence, RCS levels are crucial indicators in the progression of various pathological diseases. Besides the glyoxalase system, emerging studies report highly conserved DJ-1 super-family proteins as critical regulators of RCS. DJ-1 superfamily proteins, including the human DJ-1, a genetic determinant of Parkinson's disease, possess diverse physiological functions paramount for combating multiple stressors. Although *S. cerevisiae* retains four DJ-1 orthologs (Hsp31, Hsp32, Hsp33, and Hsp34), their physiological relevance and collective requirement remain obscure. Here, we report for the first time that the yeast DJ-1 orthologs function as novel enzymes involved in the preferential scavenge of glyoxal and methylglyoxal, toxic metabolites, and genotoxic agents. Their collective loss stimulates chronic glycation of the proteome, and nucleic acids, inducing spectrum of genetic mutations and reduced mRNA translational efficiency. Furthermore, the Hsp31 paralogs efficiently repair severely glycated macromolecules derived from carbonyl modifications. Also, their absence elevates DNA damage response, making cells vulnerable to various genotoxins. Interestingly, yeast DJ-1 orthologs preserve functional mitochondrial content, maintain ATP levels, and redistribute into mitochondria to alleviate the glycation damage of macromolecules. Together, our study uncovers a novel glycation repair pathway in *S. cerevisiae* and a possible neuroprotective mechanism of how hDJ-1 confers mitochondrial health during glycation toxicity.

**\*For correspondence:**
patrick@iisc.ac.in

[†]These authors contributed equally to this work

**Competing interest:** The authors declare that no competing interests exist.

## Editor's evaluation

This paper reports the important discovery of a mechanism of cellular response to the reactive carbonyl species, which in yeast involves Hsp31, Hsp32, Hsp33, and Hsp34, all orthologs of DJ-1 protein, a determinant of Parkinson's disease. The evidence supporting the role of Hsp31-34 in preventing and repairing damage to mitochondria, proteins and nucleic acids by reactive carbonyl species is convincing. The study will be of interest to molecular cell biologists and beyond, as it helps to shed new light on the molecular basis of Parkinson's disease pathology.

## Introduction

The structural and functional integrity of nucleic acids and proteins is critical for establishing a physiological balance to maintain normal cellular health. Reactive carbonyl species (RCS) are highly toxic compounds that covalently bind and damage vital cellular components such as protein, DNA, and fatty acids (*Semchyshyn, 2014*). RCS primarily constitutes methylglyoxal (MG), glyoxal (GO), and 3-deoxyglucosone (*Semchyshyn, 2014*). The MG and GO are metabolic by-products

of reactive aldehyde and ketones, respectively (*Allaman et al., 2015*; *Lange et al., 2012*). At the basal physiological level, these dicarbonyls participate in numerous signaling processes essential for cellular development and stress response (*Akhand et al., 2001*; *Rodrigues et al., 2020*). On the contrary, MG and GO modifies the thiol and amine groups at elevated levels through the Maillard reaction, forming a myriad of irreversible intermediates followed by stable Advanced glycation end-products (AGEs; *Semchyshyn, 2014*; *Thornalley, 2008*). They preferentially form adducts with side chains of arginine, lysine, and cysteine and indiscriminately impair the structure and function of proteins. Moreover, the glycation of essential proteins, including antioxidant machinery, can alter reactive oxygen species (ROS) homeostasis, mitochondrial performance, and proteostasis (*Goudarzi et al., 2018*; *Seo et al., 2014*). Likewise, chronic exposure of nucleic acids to RCS culminates into nucleotide AGEs that promote genetic aberrations and translational defects and induce a spectrum of mutations, including transition and G: C to T: A transversions (*Murata-Kamiya et al., 1997*; *Murata-Kamiya et al., 2000*). Elevated glycation of protein and DNA has implications for several pathologies like diabetes, aging, and neurodegenerative diseases (*Fournet et al., 2018*). Therefore, regulating endogenous RCS and minimizing its detrimental effects on biomolecules is critically important. The glyoxalase system that constitutes glyoxalase I and II efficiently detoxifies MG and GO to D-lactate and glycolate by utilizing glutathione (GSH) as a cofactor (*Inoue et al., 2011*). Since excess levels of carbonyls concomitantly induce ROS, the GSH pool is rapidly depleted during the anti-oxidation process (*de Bari et al., 2020*). Hence, cells have evolved GSH independent pathway (glyoxalase III system) to maintain a healthy redox balance, primarily comprising DJ-1/ThiJ/Pfp1 superfamily proteins (*Bankapalli et al., 2015*; *Lee et al., 2012*; *Melvin et al., 2017*).

DJ-1 superfamily members are highly conserved, multi-stress responding, and ubiquitously present in most organisms (*Bandyopadhyay and Cookson, 2004*; *Wei et al., 2007*). The superfamily includes human DJ-1(hDJ-1/*PARK7*), a well-explored protein that induces a familial form of Parkinson's disease (PD) through its genetic mutations (*Bonifati et al., 2003*). Besides having various neuroprotective functions, hDJ-1 possesses methylglyoxal and glyoxalase activity that substantially modulates endogenous carbonyls (*Lee et al., 2012*). Moreover, the deletion of *E. coli* DJ-1 members exhibited alterations in the glycation of DNA and proteins (*Lee et al., 2016*; *Richarme et al., 2017*). *Saccharomyces cerevisiae* has four homologs of DJ-1, namely Hsp31, Hsp32, Hsp33, and Hsp34, representing Hsp31 mini-family proteins (*Wilson et al., 2004*). Interestingly, Hsp32, Hsp33, and Hsp34 share ~99.5% sequence identity within them and ~70% sequence identity with Hsp31. All the paralogs possess a common catalytic triad constituting Cys138, His139, and Glu170. A characteristic hallmark of DJ-1 superfamily members is the redox-sensing catalytic cysteine, whose oxidation state determines the physiological function (*Wilson, 2011*). The emerging evidence on bacteria, humans, and plant DJ-1 members reveals a unique deglycase machinery that relieves the MG and GO glycation adducts on DNA and proteins (*Prasad et al., 2022*; *Richarme et al., 2017*; *Smith et al., 2022*). This repair mechanism is further appreciated in preventing protein aggregation and altering the epigenetic landscape of the chromatin (*Sharma et al., 2019*; *Zheng et al., 2019*). Some *S. cerevisiae* Hsp31 members are reported to possess crucial functions in regulating ROS, mitochondrial dynamics, and chaperone activity (*Aslam et al., 2016*; *Bankapalli et al., 2020*; *Miller-Fleming et al., 2014*; *Tsai et al., 2015*). However, the biological relevance of having multiple DJ-1 orthologs in *S. cerevisiae* is poorly understood due to the lack of *in vitro* evidence and difficulties in targeting specific paralog as they are genetically almost identical.

In the current report, we have uncovered a primary role of the Hsp31 paralogs as glyoxalases and protectants of the genome and mitochondria against RCS. By, governing endogenous dicarbonyl levels, they prevent macromolecular glycation and abrogate genetic mutations. In response to glycation stress, mRNA translation activity was significantly reduced in the absence of yeast DJ-1 members. Furthermore, the novel deglycase machinery efficiently repairs glycated DNA and proteins, substantially reverting deleterious alterations on biomolecules. We also show that Hsp31 paralogs impart enhanced mitochondrial integrity by maintaining functional mitochondria content, sustained ATP levels, and attenuating glycation stress on mitochondria through redistribution. Together, the dual role is critical in providing multifaceted protection to cellular and organellar macromolecules during persistent carbonyl stress.

## Results

### Deletion of Hsp31 paralogs aggravates carbonyl toxicity and induces proteome glycation

The DJ-1 superfamily members strongly relate to regulating dicarbonyl stress primarily through scavenging excess intracellular MG and GO (*Lee et al., 2012*; *Subedi et al., 2011*). However, despite being conserved, the biological significance of *S. cerevisiae* Hsp31 paralogs is uncharacterized in the presence of carbonyl toxicity. Although the paralogs have very similar amino acid sequences, we utilized the ease of genetic manipulation in yeast to unravel their association with the homeostasis of RCS (*Figure 1—figure supplement 1A*). Hsp31 paralogs were sequentially deleted in the WT background in multiple combinations such as Δ*hsp31* (Δ*31*) to Δ*hsp34* (Δ*34*), Δ*31Δ32* to Δ*31Δ34*, Δ*31Δ32Δ33* (Δ*T*), and Δ*31Δ32Δ33Δ34* (Δ*Q*) which simplifies the comprehensive understanding of the individual genes. As glyoxalase-1 (GLO1) extensively modulates endogenous RCS levels, Δ*glo1* strain was used for comparative analysis (*Gomes et al., 2005*). Phenotypic analyses under GO stress revealed a mild growth sensitivity in single deletion strains (*Figure 1A, compare panels glyoxal (GO) with control*), and the viability was further affected in Δ*31Δ32*, Δ*31Δ33*, and Δ*31Δ34*. Interestingly, Δ*T and* Δ*Q* displayed synergistic worsening of growth compared to other strains, inferring their additive overlapping functions during GO toxicity. Subsequently, the effect of MG stress was tested in the deletion background of Hsp31 members. Among the paralogs, Δ*31* alone displayed mild growth sensitivity, as reported earlier (*Figure 1A, compare panels methylglyoxal (MG) with control*; *Bankapalli et al., 2015*). On the other hand, additional loss of Hsp32, Hsp33, and Hsp34 in Δ*31* had no cumulative effect on viability.

To test whether the observed phenotypes are the consequence of impaired carbonyl homeostasis, protein modifications were examined in the deletion background of Hsp31 paralogs. To determine the GO and MG-induced alterations, anti-CML (carboxymethyl-lysine) and anti-MAGE (methylglyoxal AGE) antibodies were utilized to determine the intermediates of AGEs, respectively. The basal physiological glycation levels (absence of glycation stress) in Δ*Q* strain displayed an increment in GO-modified proteins compared to Δ*glo1* strain suggesting Hsp31 paralogs are essential for the RCS detoxification *in vivo* (*Figure 1—figure supplement 1B, C*). However, such basal glycation levels are tolerable to the cells without exhibiting growth sensitivity. Intriguingly, upon subjecting to additional external glycation stress, a significant elevation of CML levels was observed in the single mutants compared to WT (*Figure 1B*). At the same time, the deletion of additional members substantially further exacerbated the modifications. It cumulatively enhanced the protein glycation levels to a greater extent in Δ*T and* Δ*Q* than Δ*glo1*, consistent with growth sensitivity (*Figure 1C*). In contrast, the effect of MG-induced glycation was restricted to Δ*31* with a subtle increment of MAGE levels compared to WT, consistent with growth sensitivity in the presence of MG (*Figure 1D and compare growth in the last panels, Figure 1A*). Intriguingly, the collective loss of the members did not exhibit additional MAGE modification in agreement with the growth sensitivity in the presence of MG (*Figure 1E and compare growth in the last panels, Figure 1A*). This preliminary data demonstrate that yeast DJ-1 members act on specific RCS and collectively provide superior tolerance towards GO toxicity than the GLO1. Strikingly, Hsp31 alone imparts resistance for both GO and MG, preventing the persistent ongoing protein glycation.

### The absence of Hsp31 paralogs enhances nucleic acid glycation and abates mRNA translation efficiency

The nucleic acids are highly susceptible to modifications and are constantly challenged with several metabolic by-products that elicit mutations and chromosomal aberrations (*Chatterjee and Walker, 2017*). Occasionally, the alterations are irreversible and may not be repaired despite stringent repair systems, culminating to disease conditions. Since the absence of Hsp31 paralogs augments protein glycation, the glycation status of cellular nucleic acids was also assessed. Despite the absence of GO stress, DNA glycation levels were elevated in both Δ*T* and Δ*Q* (*Figure 2—figure supplement 1A, B*). However, in the presence of GO stress, the single deletion strains exhibit further enhancement in the glycation compared to WT (*Figure 2A*). At the same time, the additional loss of Hsp31 paralogs led to a synergistic increment of DNA adducts, most evident in Δ*T* and Δ*Q* strains consistent with the growth sensitivity (*Figure 2B*).

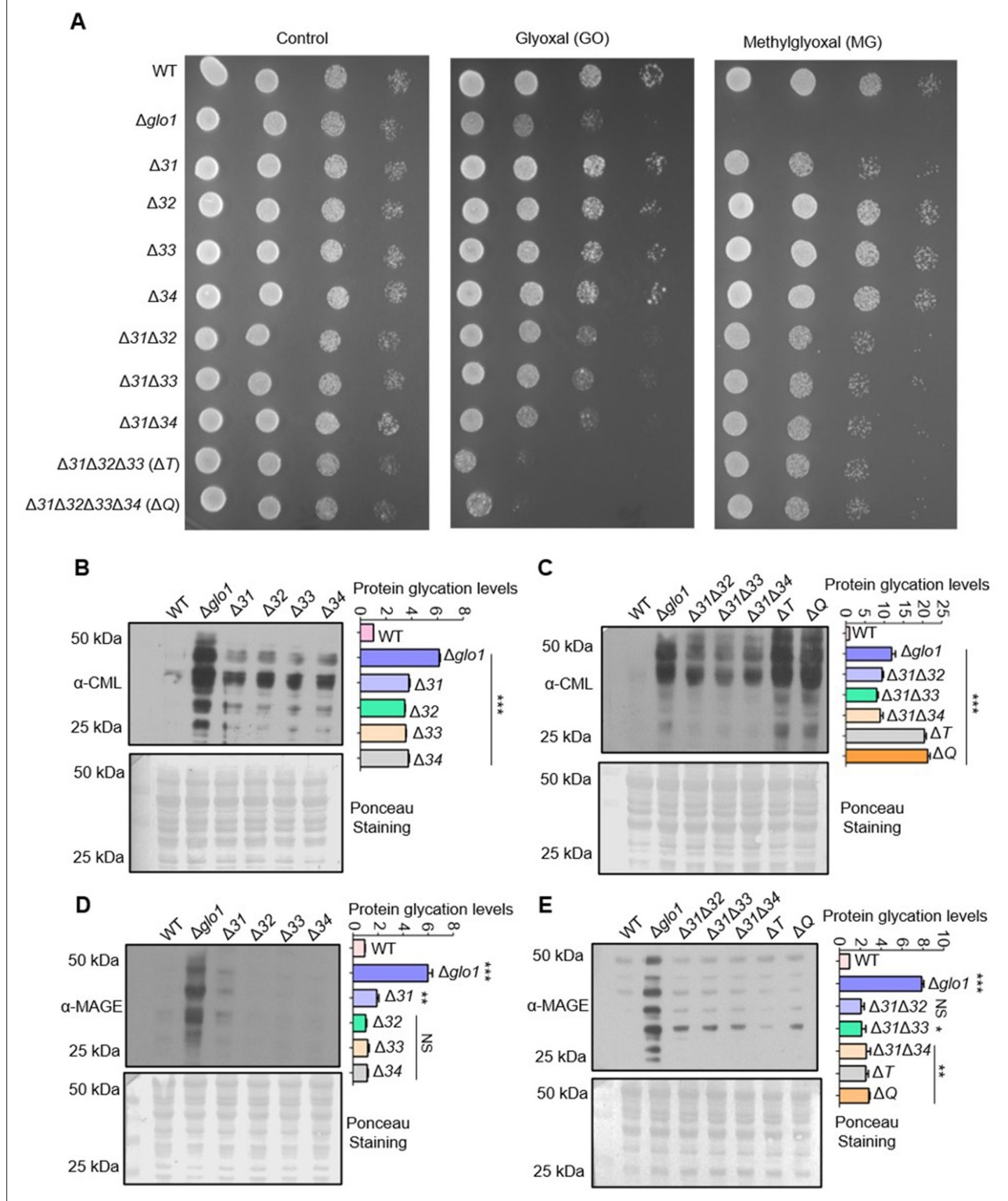

**Figure 1.** Deletion of Hsp31 paralogs induces protein glycation. (**A**) Yeast phenotypic analysis. Cells were grown until the mid-log phase and harvested, subsequently treated with 10 mM MG before being spotted or spotted on YPD medium plates containing 15 mM GO. The plates were incubated at 30 °C and imaged at 36 hr. (**B, C**) Proteome glycation profile. Yeast strains were treated with 15 mM GO in the YPD culture medium and allowed to grow for 12 hr, followed by western analysis with anti-CML antibody. (**D, E**) MAGE detection by western blotting. Cells from the mid-log phase were incubated with 10 mM MG for 12 hr, and MAGE levels were estimated using an anti-MAGE antibody. Protein glycation levels were determined by measuring the whole lane intensities by densitometry and plotted with respect to WT. The blots stained with Ponceau S were used as the loading control. One-way

*Figure 1 continued on next page*

*Figure 1 continued*

ANOVA with Dunnet's multiple comparisons test was used to determine significance from three independent biological replicates, *, p≤0.05; **, p≤0.01; ***, p≤0.001; NS, not significant.

The online version of this article includes the following source data and figure supplement(s) for figure 1:

**Source data 1.** Source data for *Figure 1B* contains raw image of western blot and densitometric values for graph.

**Source data 2.** Source data for *Figure 1C* contains raw image of western blot and densitometric values for graph.

**Source data 3.** Source data for *Figure 1D* contains raw image of western blot and densitometric values for graph.

**Source data 4.** Source data for *Figure 1E* contains raw image of western blot and densitometric values for graph.

**Figure supplement 1.** Amino acid sequence alignment of Hsp31 paralogs and protein glycation profile in the absence of GO stress.

**Figure supplement 1—source data 1.** Source data for *Figure 1—figure supplement 1B* contains raw image of western blot.

**Figure supplement 1—source data 2.** Source data for *Figure 1—figure supplement 1C* contains raw image of western blot.

Besides the alterations in DNA, the glycation of RNA was also determined, as its modifications lead to ribosome stalling, translational defects, and inefficient binding with proteins (*Mitchell et al., 2018*; *Zheng et al., 2020*). Like DNA, the lack of Hsp31 paralogs led to the high frequency of RNA modifications, which were quantitatively higher than Δ*glo1* (*Figure 2C and D*). Furthermore, the effect of glycation toxicity on global mRNA translational activity was evaluated between WT and ΔQ. In the absence of GO stress, WT and ΔQ showed similar polysome profiles (*Figure 2E*). However, the mRNA translational activity was significantly reduced in ΔQ treated with GO, as indicated by low-intensity polysome peaks (*Figure 2E*). The ratio of polysome to monosomes also suggested a reduced translation rate in ΔQ compared to WT in the presence of glycation stress (*Figure 2F*).

Next, the role of yeast DJ-1 orthologs in maintaining genome integrity was investigated during MG stress. Unlike Δ*31*, the DNA glycation in Δ*32*, Δ*33*, and Δ*34* was minimally detected by α-MAGE (*Figure 2G*), inferring their lack of participation in MG detoxification. Further, the paralogs were cumulatively deleted to evaluate the possibility of additive effects on glycation levels. Interestingly, the MAGE-modified DNA was restricted to the loss of Hsp31 due to its role in MG homeostasis (*Figure 2H*). Also, deletion of Hsp31 alone led to substantial glycation of RNA when treated with MG (*Figure 2I and J*). In summary, our results highlight that the DJ-1 paralogs in yeast play an essential role in modulating the RCS-mediated glycation of DNA and RNA.

## Yeast DJ-1 orthologs are robust glyoxalases that attenuate the glycation of macromolecules

DJ-1 superfamily members are reported to detoxify MG and GO independently by the same active site with varying kinetics (*Bankapalli et al., 2015*; *Lee et al., 2012*). Hence, to establish a direct correlation of enhanced glycation with excess intracellular RCS, Hsp31 paralogs were purified using affinity chromatography (*Figure 3—figure supplement 1A*). Our *in vitro* enzyme activity assay suggested that Hsp31, Hsp32, Hsp33, and Hsp34 possess a robust glyoxalase activity compared to BSA, which served as negative control (*Figure 3A*). Furthermore, the kinetic parameters indicate that paralogs have relative catalytic efficiencies. Notably, the paralogs exhibited fourfold higher $k_{cat}$ for glyoxalase activity than hDJ-1 (*Lee et al., 2012*; *Figure 3B* and *Figure 3—figure supplement 1B*). To further appreciate their *in vivo* biological response during GO stress, Hsp31 members were overexpressed within ΔQ strain and observed a substantial rescue in the growth similar to WT (*Figure 3C*). The expression of the proteins was normalized to equal amounts (*Figure 3—figure supplement 1C*).

To further corroborate the above findings, ΔQ strain overexpressing Hsp31 paralogs was subjected to GO stress and probed for CML levels. In agreement with our data, the modifications on proteins and nucleic acids were substantially suppressed in the presence of yeast DJ-1 glyoxalase machinery (*Figure 3D–F*). Next, the role of nucleophile cysteine amino acid (Cys-138) in the glyoxalase activity was tested, as its mutation predominantly impair the functions of DJ-1 members (*Bankapalli et al., 2015*; *Lee et al., 2012*; *Zheng et al., 2019*). Therefore, cysteine 138 was replaced with alanine (C138A), and the proteins were subjected to the glyoxalase activity assay. Upon *in vitro* analysis, all the mutant proteins had compromised activity suggesting the critical requirement of cysteine in mitigating intracellular carbonyl levels (*Figure 3G*). Furthermore, we observed that exposure of cells to excess GO concomitantly enhanced the steady-state levels of Hsp31 paralogs to approximately

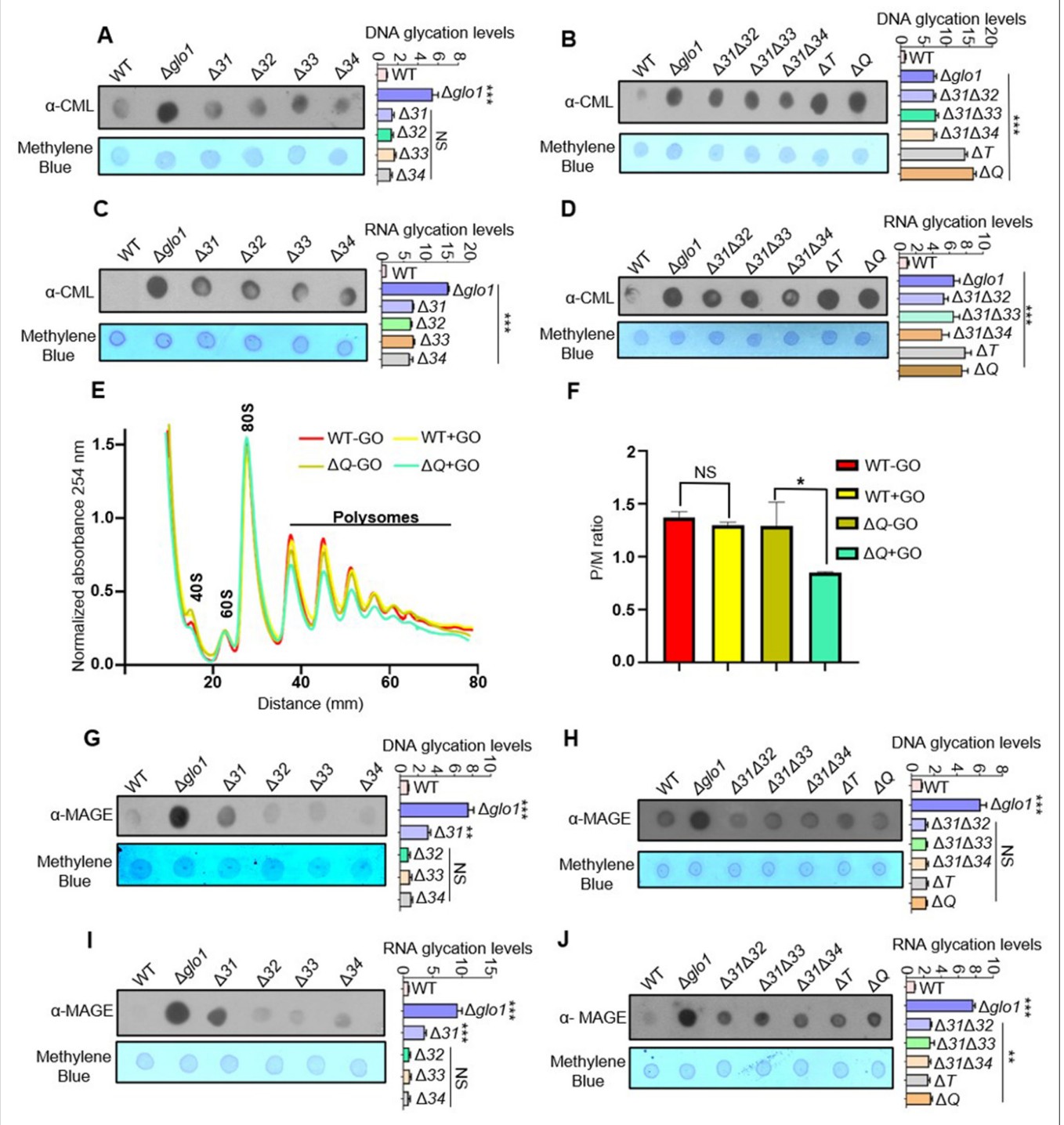

**Figure 2.** Loss of Hsp31 paralogs aggravates the glycation of DNA and RNA, affecting translational activity. (**A, B**) Immunodetection of DNA using dot-blot assay. Three µg of total genomic DNA from respective strains treated overnight with 15 mM GO was dotted on nitrocellulose membrane and probed with anti-CML antibody. (**C, D**) RNA glycation profile. Global RNA extracted from cells incubated with 15 mM GO were dotted and analyzed using anti-CML antibody. (**E**) Polysome profiling. Untreated (-GO) and treated (+GO) WT and ΔQ cells with GO were subjected to polysome profiling. (**F**) Ratio of polysomes to monosomes (P/M). The area occupied by the polysome and the monosome peak was determined using Origin 8.0 for the respective samples, and the ratios were plotted. WT (+GO) was compared with WT (-GO), and ΔQ (-GO) was compared with ΔQ (+GO). (**G–J**) Estimation of MAGE-modified nucleic acids. Strains lacking Hsp31 paralogs were treated with 10 mM MG for 12 hr, and the genomic DNA and RNA were dotted on the membrane and probed with anti-MAGE antibody. The relative intensity of dots representing the glycation levels was calculated and compared

*Figure 2 continued on next page*

*Figure 2 continued*

to WT. One-way ANOVA with Dunnet's multiple comparisons test was used to determine significance from three independent biological replicates, *, p≤0.05; **, p≤0.01; ***, p≤0.001; NS, not significant.

The online version of this article includes the following source data and figure supplement(s) for figure 2:

**Source data 1.** Source data for *Figure 2A* contains raw image of western blot and densitometric values for graph.

**Source data 2.** Source data for *Figure 2B* contains raw image of western blot and densitometric values for graph.

**Source data 3.** Source data for *Figure 2C* contains raw image of western blot and densitometric values for graph.

**Source data 4.** Source data for *Figure 2D* contains raw image of western blot and densitometric values for graph.

**Source data 5.** Source data for *Figure 2E* contains densitometric values for graph.

**Source data 6.** Source data for *Figure 2F* contains densitometric values for graph.

**Source data 7.** Source data for *Figure 2G* contains raw image of western blot and densitometric values for graph.

**Source data 8.** Source data for *Figure 2H* contains raw image of western blot and densitometric values for graph.

**Source data 9.** Source data for *Figure 2I* contains raw image of western blot and densitometric values for graph.

**Source data 10.** Source data for *Figure 2J* contains raw image of western blot and densitometric values for graph.

**Figure supplement 1.** Glycation profile of DNA in the absence of GO stress.

**Figure supplement 1—source data 1.** Source data for *Figure 2—figure supplement 1A* contains raw image of western blot.

**Figure supplement 1—source data 2.** Source data for *Figure 2—figure supplement 1B* contains raw image of western blot.

three-fold (*Figure 3—figure supplement 1D, E*). These results indicate a direct role of yeast DJ-1 orthologs in modulating CML levels through regulating endogenous glyoxal.

## Hsp31 alone specifically prevents the accretion of methylglyoxal-derived AGEs

MG is the most reactive dicarbonyl with the highest potency for glycating vital macromolecules, causing organellar damage due to membrane permeability (*Allaman et al., 2015*). To overcome MG toxicity, the glyoxalase system (GLO1 and GLO2) inevitably exploits the GSH pool for detoxification and thus alters the cellular redox balance (*de Bari et al., 2020*). On the other hand, DJ-1 members are functionally independent and do not utilize GSH as a cofactor for detoxification. To determine the methylglyoxalase activity of Hsp31 paralogs *in vitro*, we purified proteins recombinantly and subjected them to activity analysis. Strikingly, Hsp31 alone exhibited robust methyglyoxalase activity compared to other paralogs (*Figure 4A*). This further confirms the specific enrichment of MAGE-modified macromolecules found earlier in Δ*31* strain (*Figures 1D, 2G and I*). On the contrary, Hsp32, Hsp33, and Hsp34 demonstrated significantly lower methyglyoxalase activity in agreement with the lack of MAGE modifications (*Figures 1D, 2G and I*).

Remarkably, the overexpression of Hsp31 in ΔQ strain facilitated improved resistance towards MG treatment than its paralogs due to its potent methyglyoxalase activity (*Figure 4B*). Further, examining the proteome glycation by α-MAGE also indicated that Hsp31 markedly reduced MG glycated proteins compared to its paralogs (*Figure 4C*). To further substantiate the implications of methyl-glyoxalase activity of Hsp31 paralogs, modification of nucleic acids from the respective strains were investigated. Due to high intrinsic methylglyoxalase activity, Hsp31 alone could scavenge MG-associated glycation of nuclear DNA and global RNA (*Figure 4D and E*). Besides, the role of a catalytic cysteine (C138) residue in methylglyoxalase activity was evaluated due to its pivotal role in abating GO toxicity. The mutation of C138 to alanine in Hsp31 completely abolished the enzymatic activity (*Figure 4F*), which further emphasizes the prominence of a conserved cysteine in the catalytic triad of DJ-1 superfamily members. Together, our data suggest that Hsp31 provides glycation defense against both MG and GO, unlike Hsp32, Hsp33, and Hsp34.

## Hsp31 paralogs repair glyoxal glycated DNA and proteins, suppress genetic mutations, and combat genotoxic stress

Glycation of macromolecules in *S. cerevisiae* is a rapid and spontaneous event with specific targets allowing them to tune their physiological functions (*Gomes et al., 2005*). However, glycation is also an irreversible process that may erroneously alter biomolecule features without a dedicated restoration

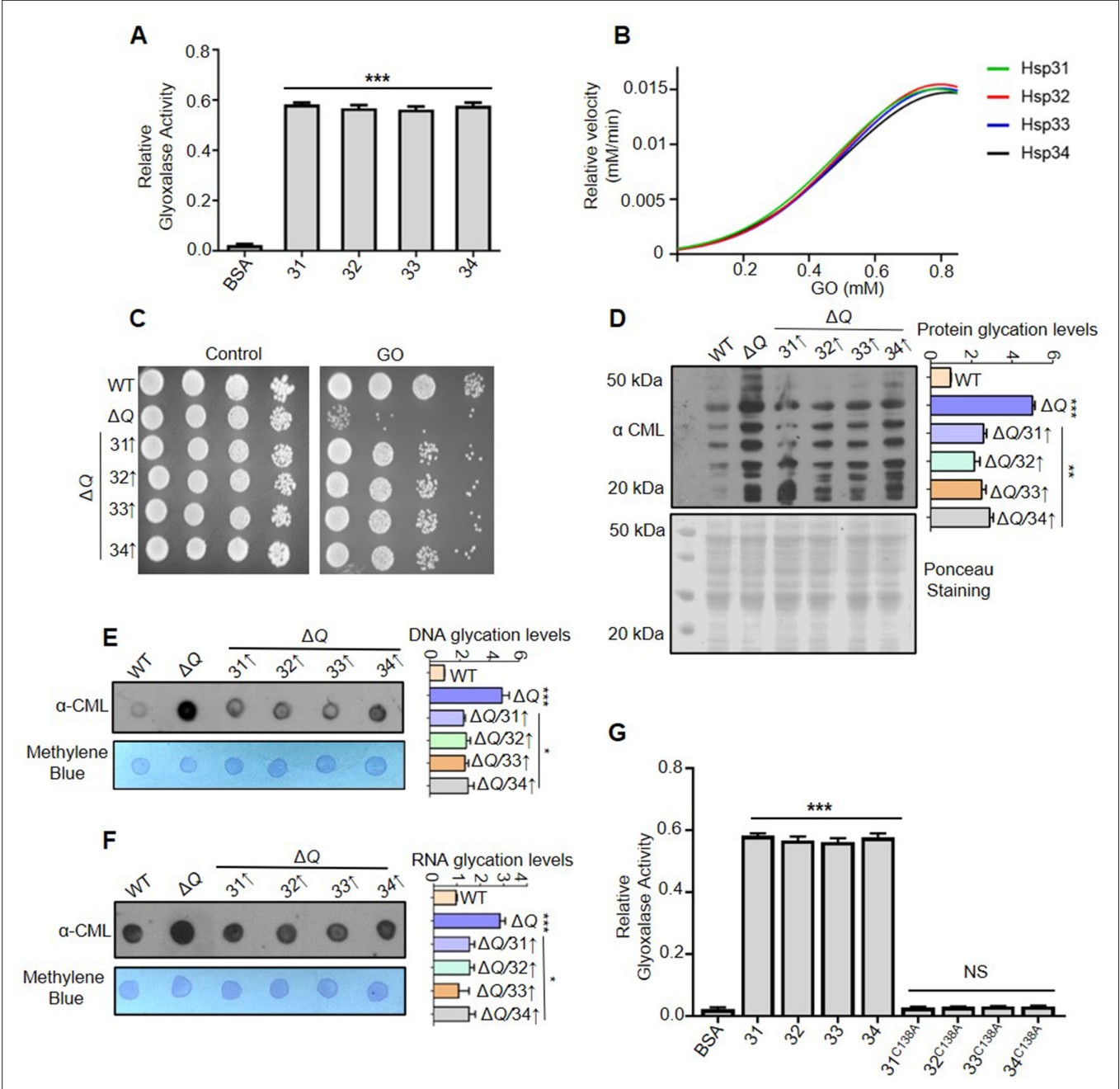

**Figure 3.** Hsp31 paralogs prevent glyoxal toxicity. (**A**) Measurement of the glyoxalase activity *in vitro*. 5 µg of purified Hsp31 paralogs were incubated with 0.5 mM GO, and the absorbance was monitored at 570 nm. (**B**) Estimating the kinetic parameters of the enzyme. The kinetic efficiency was determined by calculating their relative velocities against multiple GO concentrations. (**C**) Growth phenotypic analysis. WT and ΔQ overexpressing Hsp31 paralogs were grown until the mid-log phase, spotted on SD Leu- media plates containing 15 mM GO, and incubated at 30 °C. (**D**) Yeast DJ-1 homologs alleviate proteome glycation. Respective strains were grown in SD Leu- culture tubes and treated with 15 mM GO for 12 hr, and the whole cell lysates were processed and probed for CML modifications. (**E, F**) Analysis of nucleic acid modifications by dot-blot assay. Genomic DNA and global RNA extracted from cells treated with 15 mM GO were dotted on the membrane and detected with anti-CML antibody. (**G**) C138 amino acid residue is essential for glyoxalase activity. All the proteins were purified and incubated (5 µg) with 0.5 mM GO; subsequently, the absorbance was monitored at 570 nm. BSA was used as negative control in enzymatic assay. Glycation levels represent the relative intensities of the dots and lanes compared with WT. Data from three independent biological replicates was used to determine significance through one-way ANOVA with Dunnet's multiple comparisons test *, $p \leq 0.05$; **, $p \leq 0.01$; ***, $p \leq 0.001$; NS, not significant.

The online version of this article includes the following source data and figure supplement(s) for figure 3:

**Source data 1.** Source data for *Figure 3A* contains densitometric values for graph.

*Figure 3 continued*

**Source data 2.** Source data for *Figure 3B* contains densitometric values for graph.

**Source data 3.** Source data for *Figure 3D* contains raw image of western blot and densitometric values for graph.

**Source data 4.** Source data for *Figure 3E* contains raw image of western blot and densitometric values for graph.

**Source data 5.** Source data for *Figure 3F* contains raw image of western blot and densitometric values for graph.

**Source data 6.** Source data for *Figure 3G* contains densitometric values for graph.

**Figure supplement 1.** Enzyme kinetics of yeast DJ-1 orthologs and their expression under GO stress.

**Figure supplement 1—source data 1.** Source data for *Figure 3—figure supplement 1A* contains raw image of SDS-PAGE.

**Figure supplement 1—source data 2.** Source data for *Figure 3—figure supplement 1C* contains raw image of western blot.

**Figure supplement 1—source data 3.** Source data for *Figure 3—figure supplement 1D* contains raw image of western blot.

**Figure supplement 1—source data 4.** Source data for *Figure 3—figure supplement 1E* contains densitometric values for graph.

pathway. Emerging studies have reported hDJ-1 as deglycase (EC 3.5.1.124) that relieves advanced glycation adducts from MG and GO-modified nucleotides and proteins (*Richarme et al., 2017*; *Richarme et al., 2015*). This observation unraveled a gateway to regulating the glycation status of essential components within cells across species.

We utilized the modified protocols to test whether the yeast Hsp31 paralogs possess deglycase activity (*Prasad et al., 2022*; *Richarme et al., 2017*). Briefly, deoxyribonucleotide triphosphate (dNTPs) were pre-treated without or with GO to glycate the substrates completely. The glycation mixture was further diluted to 6-fold to suppress the effect of free GO on the activity of Hsp31 paralogs. Later, the glycated substrates were incubated for 3 hr with purified Hsp31 paralogs and BSA as a control to test deglycase activity (*Experimental scheme*; *Figure 5A*). The deglycase activity was ascertained by amplifying the *S. cerevisiae sod1* gene generated by Phusion high fidelity DNA polymerase that incorporates repaired dNTPs during PCR reaction. Unlike GO and BSA controls, incubation with Hsp31 paralogs efficiently reverted the glycation adducts of dNTPs, noted by robust amplification of the PCR product (*Figure 5B*). As an additional substrate of DNA, the forward and reverse oligonucleotide primers of the *sod1* gene were glycated in the presence or absence of GO. Besides dNTPs, yeast DJ-1 orthologs prominently deglycated the primers, allowing the reaction to yield an intense PCR product in contrast to GO and BSA controls (*Figure 5C*).

The chronic glycation of proteins induces multifactorial changes that potentially trigger aggregation, cross-linking, and cytotoxicity. Therefore, it is critical to study whether Hsp31 paralogs confer the glycation repair of proteins. To determine the deglycase property, human Sod1, and Lysozyme were pre-treated without or with GO separately for 2 hr to form glycation adducts. The reaction mixture was diluted 6-fold and incubated for 3 hr with purified Hsp31 members or BSA (negative control) (*Experimental scheme*; *Figure 5A*). The samples were separated on SDS-PAGE and subjected to immunodetection by anti-CML antibody to confirm the degree of glycation. All the Hsp31 members exhibited robust deglycase activity, as the paralogs extensively reverted the glycation damage of proteins, unlike GO and BSA controls (*Figure 5D and E*). On the other hand, the active site mutation (C138A) in paralogs was tested to verify deglycase enzyme activity further. The active site mutants failed to relieve the glycation adducts on DNA and proteins, thus establishing the specificity of the deglycase action of Hsp31 paralogs (*Figure 5—figure supplement 1A–D*).

As GO is known to induce random mutations that severely impair genome integrity, we scored the glycation-induced conversions at the DNA level to investigate the impact of glyoxalase and deglycase activity of Hsp31 paralogs on genome protection. Strains lacking Hsp31 paralogs were subjected to GO treatment, and mutation frequency was determined by amplifying specific genes such as mitochondrial-encoded (*cox2*) and nuclear-encoded (*sod1* and *rad14*), which have vital *in vivo* functions. Due to the loss of Hsp31 paralogs, the ΔQ strain exhibited enhanced mutation frequency as determined by cox2, sod1, and rad14 gene sequencing (*Figure 5F*). However, due to robust intrinsic glyoxalase and deglycase activity associated with the paralogs, the overexpression of Hsp31, Hsp32, Hsp33, and Hsp34 significantly attenuated the mutation frequency of all the genes tested, further highlighting their physiological role in the maintenance of genome integrity under the GO toxicity (*Figure 5F* and *Figure 5—figure supplement 1E*).

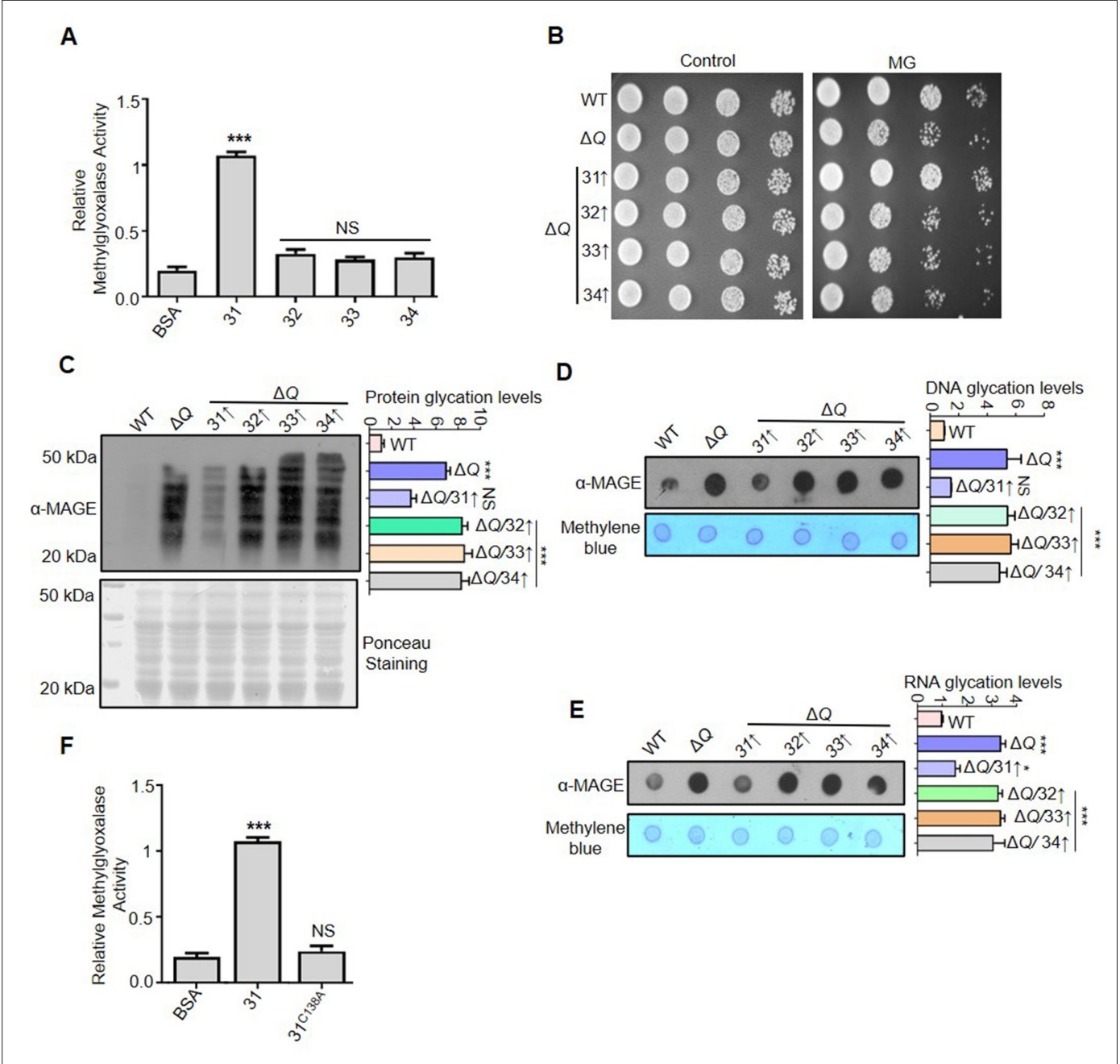

**Figure 4.** Hsp31 is a potent scavenger of methylglyoxal. (**A**) *In vitro* methylglyoxalase activity. Purified Hsp31 members 5 µg each were incubated with 0.5 mM MG, and the absorbance at 530 nm was monitored. (**B**) Spot assay. Respective strains grown till the mid-log phase were harvested and treated with 10 mM MG for 5 hr before spotting on SD Leu- plates. Images were captured at 36 hr. (**C–E**) Hsp31 reduces MG-derived AGE modifications. Individual strains were treated with 10 mM MG in the culture tubes for 12 hr, and the macromolecule glycation was analyzed using an anti-MAGE antibody. (**F**) C138 amino acid residue is critical for methylglyoxalase activity. 5 µg of Hsp31-WT and Hsp31$^{C138A}$ mutant were incubated with MG, and the activity was examined at 530 nm. BSA was used as negative control in enzymatic assay. Glycation levels represent the relative intensities of the dots and lanes compared with WT. One-way ANOVA with Dunnett's multiple comparisons test was used to determine significance from three independent biological replicates, *, $p \leq 0.05$; **, $p \leq 0.01$; ***, $p \leq 0.001$; NS, not significant.

The online version of this article includes the following source data for figure 4:

**Source data 1.** Source data for *Figure 4A* contains densitometric values for graph.

**Source data 2.** Source data for *Figure 4C* contains raw image of western blot and densitometric values for graph.

**Source data 3.** Source data for *Figure 4D* contains raw image of western blot and densitometric values for graph.

*Figure 4 continued on next page*

*Figure 4 continued*

**Source data 4.** Source data for *Figure 4E* contains raw image of western blot and densitometric values for graph.

**Source data 5.** Source data for *Figure 4F* densitometric values for graph.

To further establish their role in offering genome integrity, WT and ΔQ strains were exposed to well-known genotoxic agents such as methyl methanesulfonate (MMS) and hydroxyurea (HU). The ΔQ strain showed severe growth sensitivity towards MMS and HU treatment compared to WT (*Figure 5—figure supplement 1F*). Furthermore, treatment with GO and genotoxins combined showed a synergistic increment in the growth sensitivity of ΔQ compared to their individual treatments (*Figure 5G*). These genotoxins induce DNA lesions and strand breaks, which could be monitored by the expression of DNA damage response genes such as RNR3 (subunit of RiboNucleotide Reductase) that specifically upregulates during genotoxic stress (*Fu et al., 2008*). Intriguingly, ΔQ exhibited a notable increase in the expression of RNR3 compared to WT cells treated with MMS and GO (*Figure 5H*). Besides RNR3, multiple DNA repair proteins, including RAD52, assemble at the strand break regions. Hence, the formation of RAD52-GFP foci indicates the extent of DNA damage in cells (*Conde and San-Segundo, 2008*). Intriguingly, ΔQ strain displayed a threefold increment in RAD 52 foci formation compared to WT upon treatment with MMS and GO (*Figure 5I* and *Figure 5—figure supplement 1G*). In summary, our findings indicate that the Hsp31 members collectively attenuate the genotoxic damage and participate in an antiglycation pathway that globally detects and restores the GO-modified biomolecules.

## Hsp31 alone explicitly reverts methylglyoxal glycated macromolecules and attenuates mutation frequency

Balancing the physiological amount of dicarbonyls is a major biological challenge as they have a short half-life and quickly react with nearby biomolecules. Besides glyoxalase-I and II systems, very few enzymes have been identified to detoxifying MG (*Vander Jagt and Hunsaker, 2003*). About 90–99% of cellular MG is bound to macromolecules, which may be reversible by the action of deglycases (*Allaman et al., 2015*). We utilized a modified protocol to test whether the paralogs attenuate MG-mediated modification of DNA and proteins (*Richarme et al., 2017*). As illustrated in schematic *Figure 6A*, the dNTPs or DNA oligonucleotide primers substrates were incubated in the absence or presence of MG, followed by sixfold dilution of the reaction mixture to minimize the effects of free MG on the enzymatic activity of Hsp31 paralogs. Later, purified DJ-1 members were added, and deglycase activity was ascertained by PCR of the *sod1* gene using Phusion high-fidelity DNA polymerase. In line with our findings, Hsp31 alone suppressed the MG-mediated glycation through its deglycase activity, yielding robust amplification of PCR product (*Figure 6B and C*).

To delineate the MG-associated protein deglycation, hSod1, and Lysozyme were treated without or with MG and probed with α-MAGE antibody. In the presence of Hsp31, the glycation adducts on the proteins was significantly reverted (*Figure 6D and E*). On the other hand, the Hsp32, Hsp33, and Hsp34 failed to revert the glycation of proteins *in vitro*, similar to DNA substrates (*Figure 6D and E*). The Hsp31 active site mutant (C138A) also failed to repress the MG adducts on DNA and proteins, underlining its significance in the catalytic reaction (*Figure 6—figure supplement 1A–D*). We also determined if the dual role of Hsp31 could provide genome integrity by scoring the MG-induced DNA mutations. Strikingly, the ΔQ strain displayed an augmented mutation rate, which was notably abated by the expression of Hsp31 during MG toxicity, confirmed by amplification and sequencing of selected yeast-specific genes (*Figure 6F* and *Figure 6—figure supplement 1E*). Thus, our data establish a critical function of Hsp31 in suppressing the MG-mediated aberrant glycation of macromolecules in *S. cerevisiae*.

## Dicarbonyl stress induces translocation of Hsp31 paralogs into mitochondria for organelle protection

DJ-1 superfamily members display dynamic subcellular localization into various organelles, such as the nucleus and mitochondria, during the stress response (*Bankapalli et al., 2015*; *Conde and San-Segundo, 2008*; *Fu et al., 2008*). However, the significance of the redistribution is still unclear. Therefore, we investigated whether Hsp31 paralogs localize to mitochondria during GO toxicity. To determine, the Hsp31 paralogs were genomically fused with GFP, and the mitochondria were

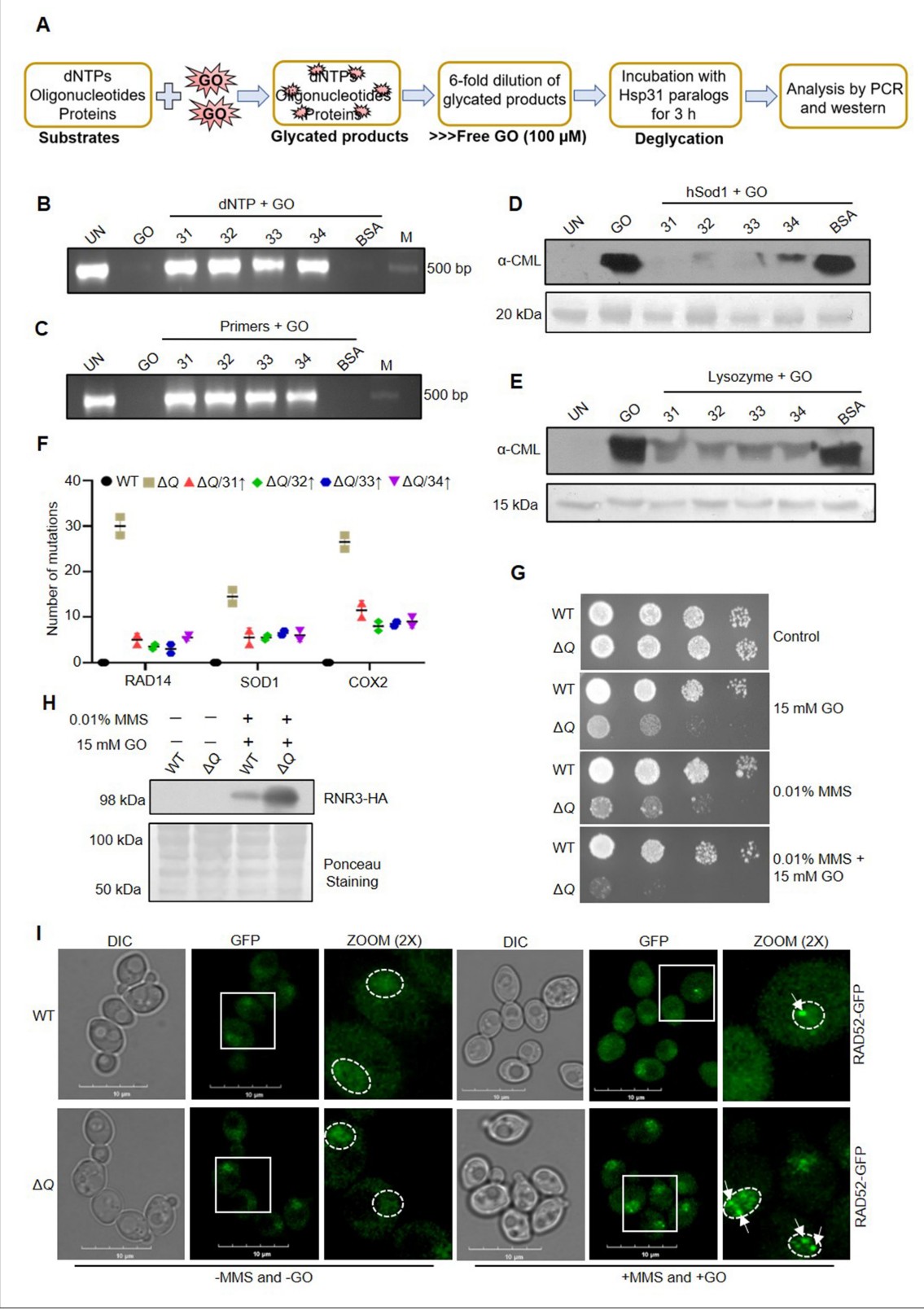

**Figure 5.** *In vitro* deglycation of DNA and proteins, and genotoxic sensitivity in the absence of yeast DJ-1 orthologs. (**A**) Schematic representation of the experimental procedure followed for DNA and protein deglycation reactions. (**B, C**) Hsp31 paralogs deglycate DNA. 500 µM dNTPs (**B**) or 150 µM forward and reverse primers (**C**) were incubated without (UN) or with 2 mM GO for 2 hr, followed by 3 hr incubation with 5 µg Hsp31 paralogs. The samples were examined for deglycation through PCR. (**D, E**) Yeast DJ-1 members repair glycated proteins. 2 µg of purified protein hSod1 (**D**) or

*Figure 5 continued on next page*

*Figure 5 continued*

Lysozyme (**E**) were treated without (UN) or with 2 mM GO for 2 hr, and the reactions were further incubated for 3 hr with Hsp31 paralogs. Anti-CML antibody was used to determine glycation levels. BSA was used as a negative control in all experiments. 10 kb DNA ladder was used as a marker (**M**) for DNA gels, and the Ponceau S stain indicates the equal loading of protein samples. (**F**) Hsp31 paralogs attenuate genetic mutations. Individual genes were PCR amplified, and Sanger sequenced from the isolated genome of strains treated with 15 mM GO. The number of genetic mutations was calculated and plotted on GraphPad prism 5.0 (n=2). (**G**) Growth phenotypic analysis. Cells grown until the mid-log phase were spotted on plates containing 0.01% MMS or 15 mM GO or 0.01% MMS and 15 mM GO. The plates were incubated at 30 °C and imaged at 36 hr. (**H**) Western analysis of RNR3 levels. WT and ΔQ grown till the mid-log phase were supplemented with 0.01% MMS, and 15 mM GO in the culture media and further incubated for 3 hr. Subsequently, RNR3 levels were probed using an anti-HA antibody. (**I**) RAD52 foci formation. Cells were grown until the mid-exponential phase and were treated without (-) or with 0.03% MMS and 15 mM GO for 1 hr. Subsequently, cells were imaged using a confocal microscope (Olympus FV3000). Representative images have a 10 µm scale. All the experiments were performed in three independent biological replicates.

The online version of this article includes the following source data and figure supplement(s) for figure 5:

**Source data 1.** Source data for *Figure 5B* contains raw image of agarose gel.

**Source data 2.** Source data for *Figure 5C* contains raw image of agarose gel.

**Source data 3.** Source data for *Figure 5D* contains raw image of western blot.

**Source data 4.** Source data for *Figure 5E* contains raw image of western blot.

**Source data 5.** Source data for *Figure 5H* contains raw image of western blot.

**Source data 6.** Source data for *Figure 5F* contains multiple sequence alignment highlighting the mutations.

**Figure supplement 1.** GO-associated macromolecular deglycation, mutation frequency profile, and genotoxic damage.

**Figure supplement 1—source data 1.** Source data for *Figure 5—figure supplement 1A* contains raw image of agarose gel.

**Figure supplement 1—source data 2.** Source data for *Figure 5—figure supplement 1B* contains raw image of agarose gel.

**Figure supplement 1—source data 3.** Source data for *Figure 5—figure supplement 1C* contains raw image of western blot.

**Figure supplement 1—source data 4.** Source data for *Figure 5—figure supplement 1D* contains raw image of western blot.

**Figure supplement 1—source data 5.** Source data for *Figure 5—figure supplement 1G* contains densitometric values of graph.

decorated by targeting mCherry tagged with mitochondrial targeting sequence (MTS). Under normal physiological conditions, the paralogs were predominantly localized in the cytosol (*Figure 7A–D*, *-GO untreated panels*). Strikingly, upon treatment of cells with GO, all the Hsp31 paralogs translocate into mitochondria, as highlighted by the merged images of GFP and mCherry (*Figure 7A–D*,*+GO stress panels*).

To further support the microscopic findings, western analysis was performed to quantitatively measure the mitochondrial localization of Hsp31 paralogs from fractionated mitochondria of GO-treated and untreated cells. A ~2.5-fold level increase in the localization of Hsp31 paralogs into mitochondria was observed under GO toxicity (*Figure 7—figure supplement 1A*). As Hsp31 regulates MG-induced glycation of macromolecules, we determined its redistribution in response to MG stress. The microscopic visualization suggests that Hsp31 upon MG toxicity localizes into mitochondria (*Figure 7E*), which was further quantified from probing isolated mitochondria treated with MG stress. The immunoblots indicate a 2-3 fold enrichment of Hsp31 in mitochondria upon exposure to MG (*Figure 7—figure supplement 1B*). In contrast, Hsp32, Hsp33, and Hsp34 do not display mitochondrial redistribution under MG toxicity (*Figure 7—figure supplement 1C–E*), further emphasizing that Hsp31 and not its paralogs respond and mitigate broader stress conditions. Tim23 and Ydj-1 were probed for mitochondrial and cytosol control proteins, respectively (*Figure 7—figure supplement 1F*).

To understand the relevance of GO-dependent mitochondrial translocation of Hsp31 paralogs, mitochondria were isolated from cells supplemented with GO to analyze the altered glycation of mitochondrial macromolecules. Intriguingly, a significant portion of the mitochondrial proteome and DNA was glycated in ΔQ strain, while in the presence of Hsp31 paralogs, the glycation levels were substantially suppressed (*Figure 7F and G*). Since the Hsp31 paralogs govern the glycation of mitochondrial macromolecules, we studied mitochondrial health by determining the viability of deletion strains of yeast DJ-1 members on a non-fermentable carbon source (glycerol). Although we observe no growth difference in dextrose media, the collective absence of Hsp31 paralogs in ΔT and ΔQ induced sensitivity to glycerol media (*Figure 8—figure supplement 1A*). These results suggest that the disruption of yeast DJ-1 members compromises mitochondrial health.

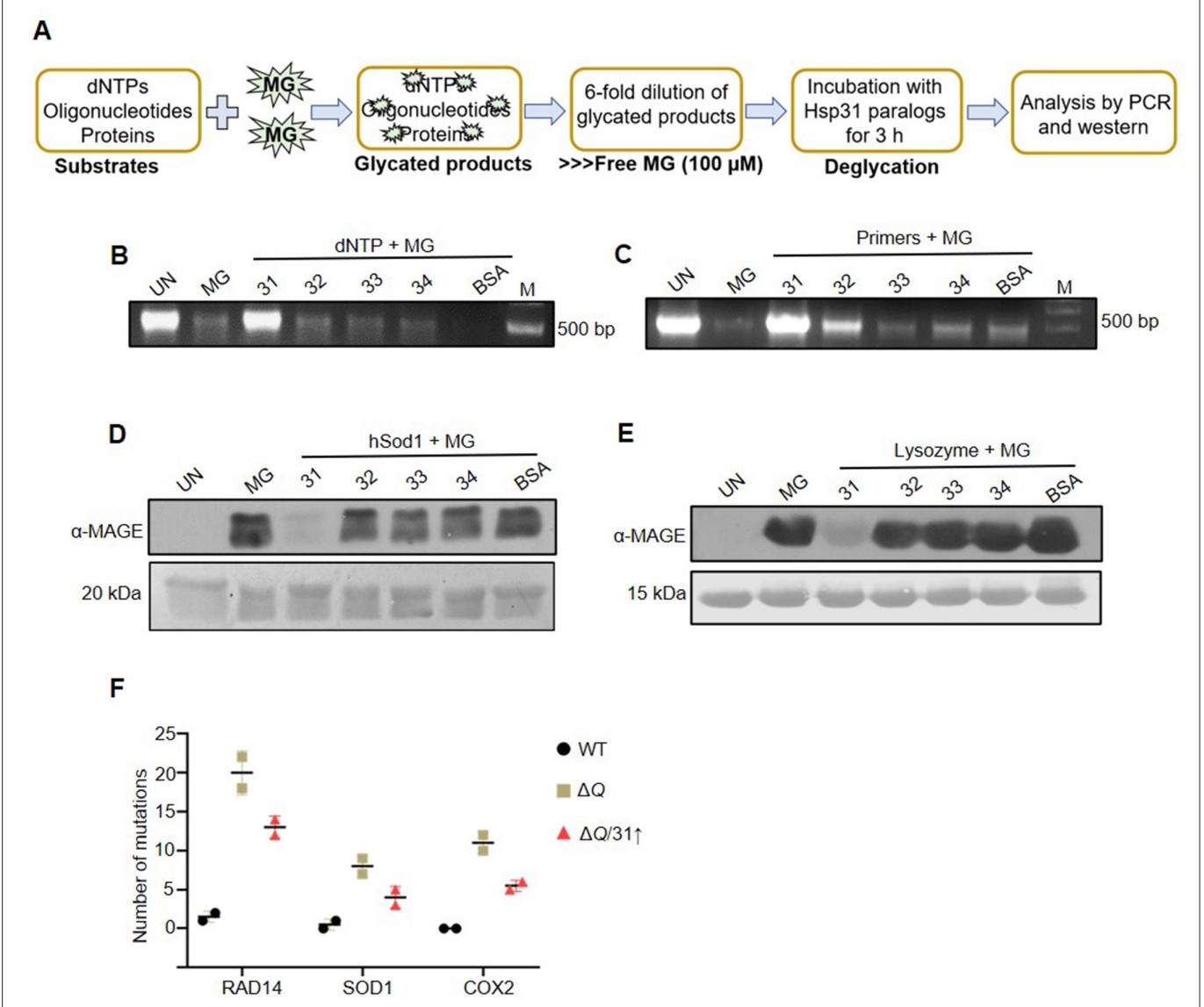

**Figure 6.** Hsp31 repairs MG-derived AGE modifications on DNA and proteins. (**A**) Schematic representation of the experimental procedure used for DNA and protein deglycation reactions. (**B, C**) DNA deglycation by Hsp31. 500 µM dNTPs (**B**) or 150 µM forward and reverse primers (**C**) were incubated without (UN) or with 2 mM MG for 2 hr. Subsequently, 5 µg Hsp31 paralogs were added and incubated for 3 hr. The samples were subjected for PCR analysis. (**D, E**) Hsp31 reverts MG modification on proteins. hSod1 (**D**) and Lysozyme (**E**) were glycated for 2 hr with 2 mM MG and incubated with Hsp31 paralogs. The glycation status was determined through western analysis against anti-MAGE antibody. 10 kb DNA ladder was used as a marker (**M**) for DNA gels, and the Ponceau S stain indicates the equal loading of protein samples. BSA was used as a negative control in all experiments. The experiments were performed in three independent biological replicates. (**F**) The dual role of Hsp31 reduces MG-induced DNA mutations. Respective genes were PCR amplified, and Sanger sequenced from strains treated with 10 mM MG. The number of genetic mutations was plotted on GraphPad prism 5.0 (n=2).

The online version of this article includes the following source data and figure supplement(s) for figure 6:

**Source data 1.** Source data for *Figure 6B* contains raw image of agarose gel.

**Source data 2.** Source data for *Figure 6C* contains raw image of agarose gel.

**Source data 3.** Source data for *Figure 6D* contains raw image of western blot.

**Source data 4.** Source data for *Figure 6E* contains raw image of western blot.

**Source data 5.** Source data for *Figure 6F* contains multiple sequence alignment highlighting the mutations.

**Figure supplement 1.** Hsp31$^{C138A}$ fails to repair glycated DNA and proteins.

**Figure supplement 1—source data 1.** Source data for *Figure 6—figure supplement 1A* contains raw image of agarose gel.

*Figure 6 continued on next page*

*Figure 6 continued*

**Figure supplement 1—source data 2.** Source data for *Figure 6—figure supplement 1B* contains raw image of agarose gel.

**Figure supplement 1—source data 3.** Source data for *Figure 6—figure supplement 1C* contains raw image of western blot.

**Figure supplement 1—source data 4.** Source data for *Figure 6—figure supplement 1D* contains raw image of western blot.

Therefore, we further explored mitochondrial maintenance by Hsp31 paralogs by investigating various health parameters of the organelle. The mitochondrial morphology indicates the dynamic balance of the fusion-fission network, which could be affected by altered mitochondrial health. Under physiological conditions, ΔQ displayed a highly reticular mitochondrial network than WT, possibly due to elevated expression of Mgm1 and Fzo1 as reported by *Bankapalli et al., 2020* in *Δ31Δ34* (*Figure 8A -GO panel*). However, under glycation stress, WT showed fragmented morphology of mitochondria as compared to no stress (*Figure 8A +GO panel*). Strikingly, despite the increase in the reticular network, we observed higher fragmentation of mitochondria in ΔQ than WT upon glycation toxicity (*Figure 8A +GO panel*). As the glycation affected the mitochondrial morphology, we examined the impact on total and functional mitochondrial content through flow cytometric analysis using NAO and TMRE dyes, respectively. In the absence of GO stress, ΔQ showed an increase in overall mitochondrial content than WT, possibly due to hyperfusion of mitochondria (*Figure 8B*). Although the mass of mitochondria remained almost similar in WT, an increase in total mitochondrial content was found in ΔQ upon GO stress (*Figure 8B*). Next, functional mitochondrial mass was determined through TMRE dye, we observed comparable functional mitochondria in WT treated and untreated, while a significant loss of functional mitochondria was noted in ΔQ under glycation toxicity (*Figure 8C*). To score the glycation-induced damage on mitochondrial health, ATP levels were estimated from isolated mitochondria. Intriguingly, the glycation stress led to a substantial loss of mitochondrial ATP levels by ~40% in ΔQ compared to WT (*Figure 8D*). These results infer that glycation stress induces extensive mitochondrial damage and impaired clearances of dysfunctional mitochondria in the absence of Hsp31 paralogs.

As mtDNA is prone to glycation in the absence of yeast DJ-1 members (*Figure 7G*), we examined the integrity of mtDNA by staining with mtDNA-specific SYTO18 stain (*Ingavale et al., 2008*; *Matta et al., 2017*). Incubation of WT with GO led to a reduction in SYTO18 staining with coalescent nucleoid morphology compared to untreated cells (*Figure 8E, left panel*). On the other hand, the lack of Hsp31 paralogs led to a significant reduction in SYTO18 staining with punctate nucleoid under GO stress compared to untreated cells (*Figure 8E, right panel*), suggesting their importance in maintaining mtDNA quantity and quality. In conclusion, the detoxification of carbonyls and macromolecular repair by yeast DJ-1 members is critical for preserving the global integrity of healthy mitochondria.

## Discussion

### Glyoxalase defence mechanism of the Hsp31 class of proteins is more robust than GLO-1 system

The present study highlights robust glyoxalase activities of yeast Hsp31 mini-family proteins essential to combat aberrant intracellular levels of RCS. Our *in vitro* experiments firmly establish the specificity of Hsp31 paralogs towards GO substrates, while Hsp31 alone detoxifies both MG and GO. Since Hsp32 and Hsp34 are structurally uncharacterized, we considered overlapping structural features with Hsp33 for comparative analysis due to its solved crystal structure and close sequence similarity. Strikingly, Hsp32, Hsp33, and Hsp34 possess a similar active site offering different substrate specificity than Hsp31. Such preferential scavenging of carbonyls is also observed in bacteria and plant DJ-1 members where YajL, ElbB, *At*DJ-1A, and *At*DJ-1B are GO specific, and some orthologs detoxify both substrates (*Kwon et al., 2013*; *Lee et al., 2016*). Although MG and GO are detoxified through distinct pathways, certain NADPH-dependent enzymes like aldehyde dehydrogenase and aldose reductase scavenge both the carbonyls, with their activities stringently governed by cellular redox status (*Vander Jagt and Hunsaker, 2003*).

In contrast to MG catabolism, the primary antioxidant GSH inefficiently reacts with GO making it a poor substrate for the GLO1 glyoxalase system (*Yang et al., 2011*). Hence, *S. cerevisiae* has bonafide multiple DJ-1 orthologs, and their steady-state levels are rapidly upregulated in response

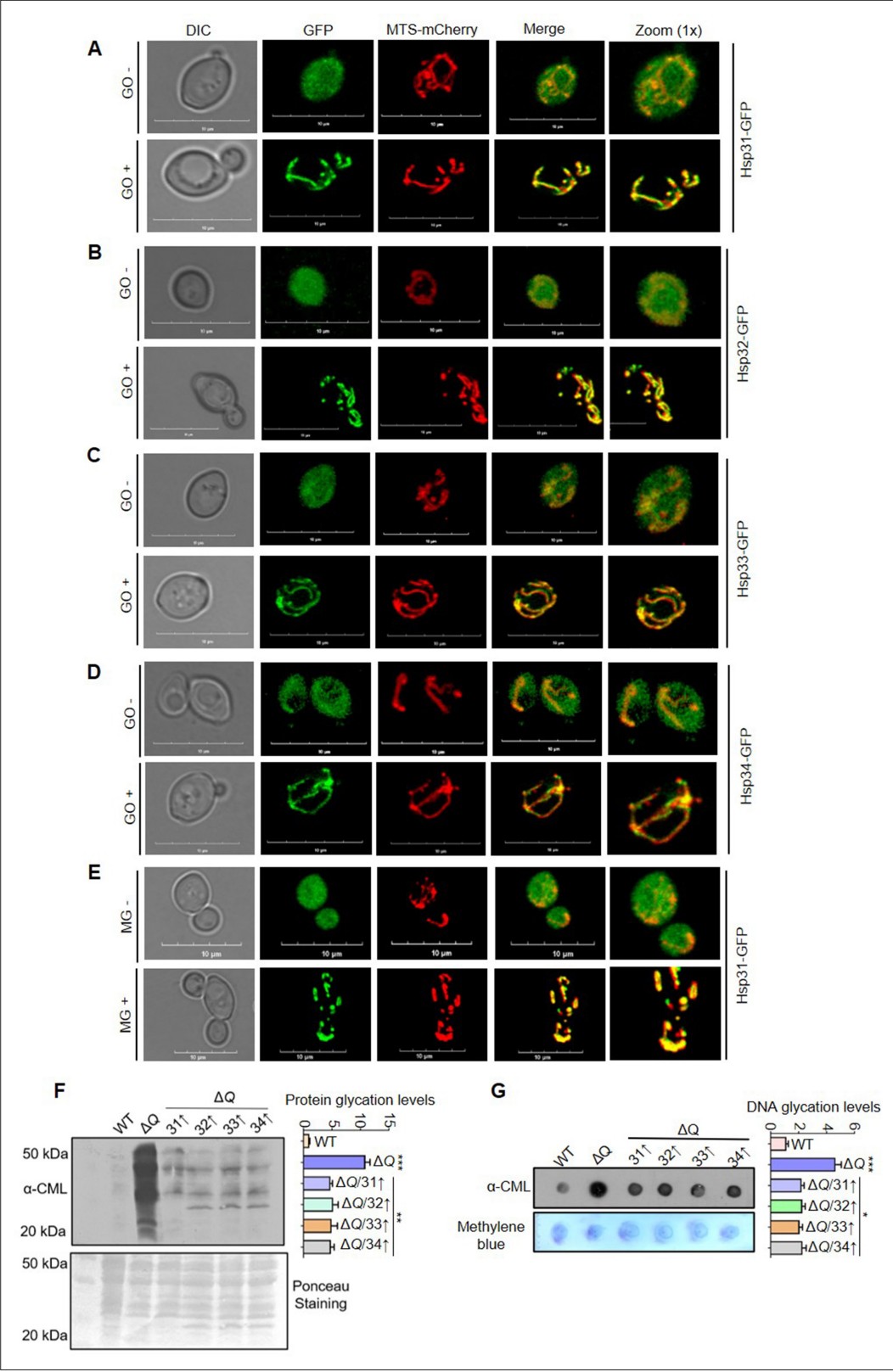

**Figure 7.** Dicarbonyl-induced translocation of yeast DJ-1 orthologs into mitochondria. (**A–E**) Mitochondrial translocation of Hsp31 paralogs. WT strain expressing genomic GFP tagged Hsp31 paralogs, and MTS-mCherry (decorates mitochondria) plasmid were treated with either buffer (-GO/-MG) or 15 mM GO (+GO) or 10 mM MG (+MG) for 3 hr. Consequently, images were captured in a confocal microscope (Olympus FV3000) and represented

*Figure 7 continued on next page*

*Figure 7 continued*

with 10 µm scale. (**F–G**) Mitochondrial protein and DNA glycation levels. WT and ΔQ overexpressing Hsp31 class of proteins were treated with 15 mM GO stress, followed by isolation of mitochondria and western analysis to determine glyoxal modifications using anti-CML antibody. The intensity of each lane and dot was quantitated densitometrically and plotted in the graph. One-way ANOVA with Dunnett's multiple comparisons test was used to determine significance from three independent biological replicates, *, $p \leq 0.05$; **, $p \leq 0.01$; ***, $p \leq 0.001$; NS, not significant.

The online version of this article includes the following source data and figure supplement(s) for figure 7:

**Source data 1.** Source data for *Figure 7F* contains raw image of western blot and densitometric values for graph.

**Source data 2.** Source data for *Figure 7G* contains raw image of western blot and densitometric values for graph.

**Figure supplement 1.** Western and microscopic analysis of mitochondrial translocated Hsp31 paralogs.

**Figure supplement 1—source data 1.** Source data for *Figure 7—figure supplement 1A* contains raw image of western blot and densitometric values for graph.

**Figure supplement 1—source data 2.** Source data for *Figure 7—figure supplement 1B* contains raw image of western blot and densitometric values for graph.

**Figure supplement 1—source data 3.** Source data for *Figure 7—figure supplement 1F* contains raw image of western blot.

to GO allowing efficient carbonyl homeostasis. Our results highlight that the deletion of individual paralogs shows moderate sensitivity for GO. However, the collective loss of yeast DJ-1 orthologs (ΔQ) renders severe growth deficiency more than Δglo1. At the same time, their absence promotes severe accumulation of toxic carbonyls leading to a higher fold macromolecular glycation than Δglo1. Besides scavenging RCS, Hsp31 is also reported to provide partial tolerance toward acetic acid stress, also a deleterious glycolytic intermediate (*Ansari et al., 2018*; *Natkańska et al., 2017*). These findings suggest that Hsp31 members are paramount in regulating a broad group of toxic metabolites compared to the GLO system (1 and 2), which is restricted to specific carbonyls. Moreover, under oxidative damage, the activity of the GLO system (1 and 2) is impaired due to extensive utilization of the limiting pool of GSH. In contrast, the GSH-independent Hsp31 paralogs aid in restoring redox balance and simultaneously provide robust RCS homeostasis, hence, indispensable for cellular viability under carbonyl stress.

## Global surveillance of nucleic acid glycation by yeast DJ-1 orthologs regulates the genome integrity

Despite several harmful effects, the physiological amounts of glycation are critical for post-translation modification, required for signaling in numerous pathways (*Akhand et al., 2001*; *Rodrigues et al., 2020*). Both MG and GO are known to glycate histone proteins, hitherto, 28 site-specific modifications on histones have been identified as targets of MG (*Ansari et al., 2018*; *Galligan et al., 2018*; *Ray et al., 2022*). Such non-enzymatic covalent epigenetic modifications regulate stress adaptations by altering chromatin architecture and gene expression. We highlight that Hsp31 paralogs are essential repair enzymes that regulate the degree of nucleic acid glycation by indirectly detoxifying the carbonyls or directly removing adducts by deglycation. This corroborates well with the idea that Hsp31 paralogs preserve genome integrity by lowering deleterious mutations as a counter-response (*Figure 9*). Also, the absence of paralogs elicits enhanced sensitivity to DNA damage stress when exposed to multiple genotoxins, as indicated by enriched expression of RNR3 and RAD52 foci formation, suggesting their strong association with genome maintenance. At the molecular level, the anti-glycation property of the paralogs could also prevent the replication stress by replenishing repaired dNTP pool. Hsp31 members are widely recognized for metabolic and transcriptional reprogramming during the diauxic shift as they extensively tune the expression of necessary proteins for rapid acclimatization (*Miller-Fleming et al., 2014*). These observations may point to a unique regulatory mechanism by Hsp31 class of proteins similar to hDJ-1 in governing the carbonyl biology of chromatin landscape (*Zheng et al., 2019*).

Besides DNA, we found enriched glycation of RNA in ΔQ due to detrimental levels of MG and GO, accompanied by the disrupted mRNA translation at the global scale, possibly through compromised binding or stalling of ribosomes. Notably, yeast DJ-1 members extensively reverted the AGE

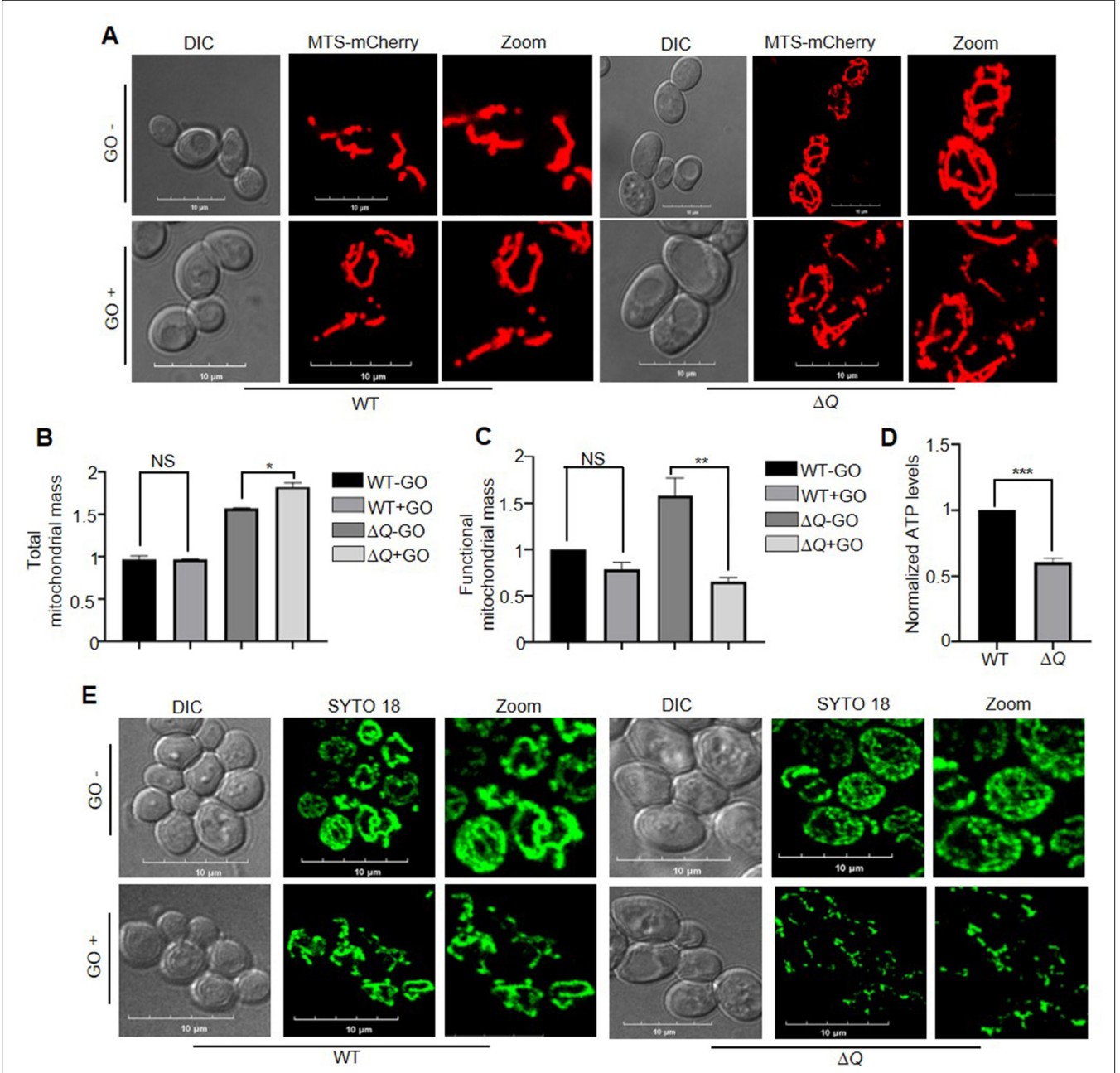

**Figure 8.** Loss of yeast DJ-1 members induces mitochondrial dysfunction. (**A**) Visualization of mitochondrial morphology. WT and ΔQ strains expressing MTS-mCherry (decorates mitochondria) were either treated with buffer (-GO) or 15 mM GO (+GO), followed by imaging. (**B, C**) FACS analysis to estimate total and functional mitochondrial mass. Respective strains were grown till the early log phase, followed by incubation with GO. Later, the cells were stained with Nonyl Acridine Orange (NAO) for total mass and TetraMethylRhodamine ethyl ester (TMRE) for determining functional mass. (**D**) Measurement of ATP levels. Selected strains were exposed to GO treatment at the mid-log phase. Consequently, the mitochondria were isolated, and the ATP levels were estimated through a fluorescence assay. (**E**) mtDNA staining by SYTO18. Following the treatment with GO, WT, and ΔQ were stained with SYTO18 dye. The microscopic analysis was performed in a confocal microscope (Olympus FV3000), with 10 μm scale in images. All experiments were performed in three independent biological replicates and analysed through paired t-test to determine significance, *, p≤0.05; **, p≤0.01; ***, p≤0.001; NS, not significant.

The online version of this article includes the following source data and figure supplement(s) for figure 8:

**Source data 1.** Source data for *Figure 8B* contains densitometric values for graph.

**Source data 2.** Source data for *Figure 8C* contains densitometric values for graph.

**Source data 3.** Source data for *Figure 8D* contains densitometric values for graph.

**Figure supplement 1.** Phenotypic analysis of Hsp31 deletion strains in non-fermentable carbon source.

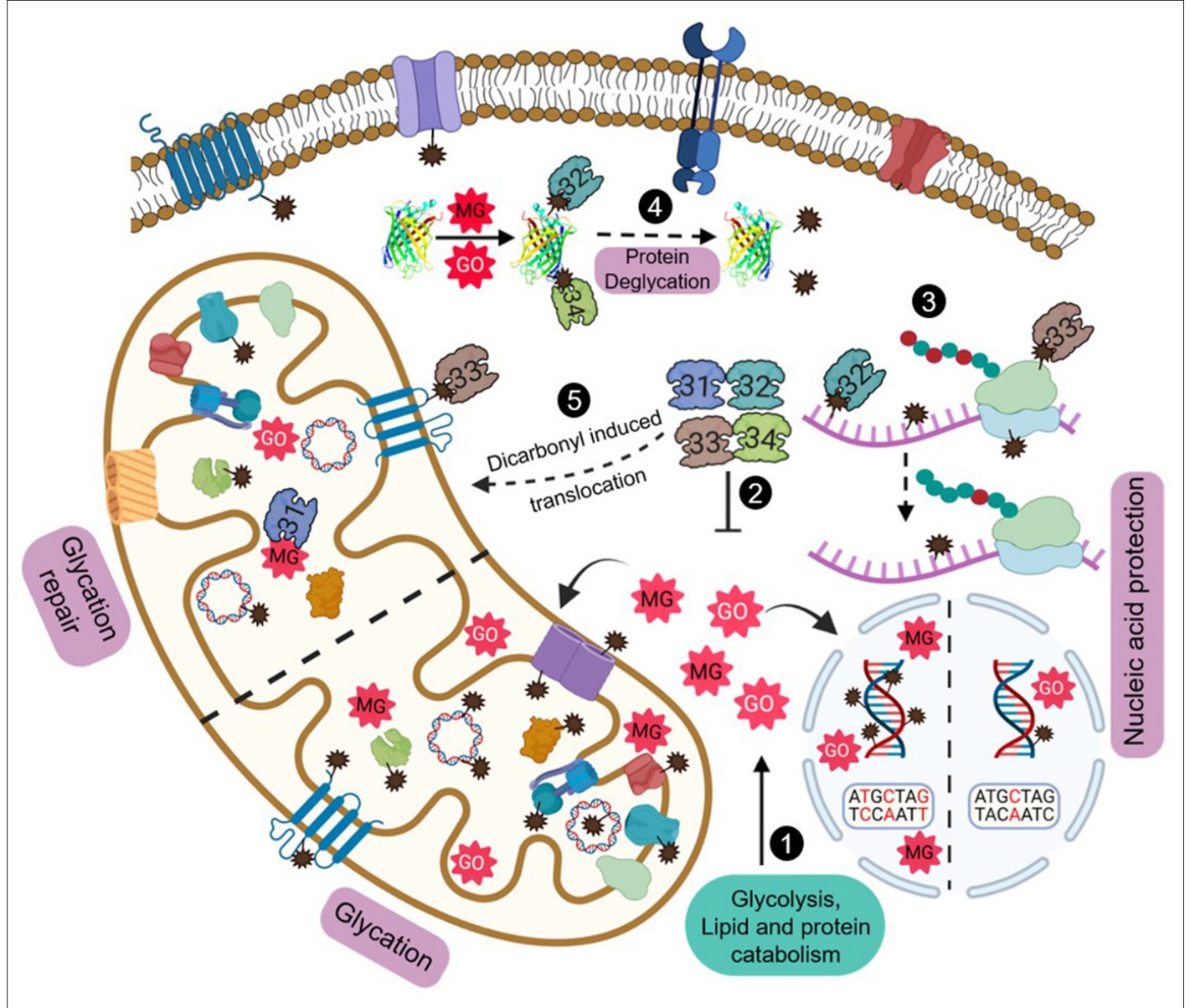

**Figure 9.** Model depicting the role of yeast DJ-1 orthologs in the maintenance and protection against carbonyls. Various metabolic pathways generate MG and GO that spontaneously glycate cytosol and organellar macromolecules (**1**). Hsp31 paralogs scavenge excess endogenous RCS and regulate their levels (**2**). The dual role of paralogs provides nucleic acid protection by efficiently detoxifing the carbonyls and reversing of damaged molecules (**3**). Deglycation of proteins preserves the functional integrity of proteins (**4**). Dicarbonyl stress-induced translocation of Hsp31 paralogs into mitochondria confers enhanced organellar protection by attenuating damage to mitochondrial biomolecules (**5**). Created with BioRender.

modifications on RNA, suggesting their comprehensive role in translation regulation. The enhanced spectrum of mutations observed in various essential genes of ΔQ is a crucial indicator of a dedicated glycation repair pathway in *S. cerevisiae* (*Figure 9*). Interestingly, a recent report on *Arabidopsis thaliana* identified a novel glycation repair pathway similar to *S. cerevisiae*, involving *At*DJ-1D in the repair of DNA and proteins (*Prasad et al., 2022*). Supporting the previous studies (*Galligan et al., 2018*; *Nair et al., 2018*; *Richarme et al., 2017*), we report yeast DJ-1 orthologs as a novel class of deglycases that repair RCS-mediated glycation of nucleic acids.

## Hsp31 paralogs regulate the proteome glycation homeostasis

Proteins are the most abundant biomolecules with considerably longer half-lives, making them highly vulnerable to multiple non-enzymatic covalent modifications. Under physiological conditions,

glycation on arginine, lysine, and cysteine modulates the functional diversity of various proteins (**Sun et al., 2019**). We show that the absence of Hsp31 paralogs triggers severe glycation of the proteome, which may contribute to protein aggregation, misfolding, and cross-linking. Furthermore, under these circumstances, the conventional proteasomal degradation pathway fails to provide quality turnover resulting in AGE accumulation (**Raupbach et al., 2020**). As a compensatory response mechanism, amino acid biosynthesis and protein translation were upregulated in the single deletions of Hsp31 mini-family proteins (**Miller-Fleming et al., 2014**). This primarily indicates the biogenesis of functional proteins to promote cell survival during glycation stress. Therefore, scavenging excess endogenous RCS by the Hsp31 paralogs is imperative to evade persistent glycation and maintain balanced proteostasis. Besides, their robust deglycase activity repairs severely glycated proteins from different cellular compartments, including mitochondria. This enables cells to efficiently reutilize the cellular proteome pool (**Figure 9**). In contrast, human DJ-1 exhibited weaker methylglyoxalase and deglycase activity than yeast and plant counterparts (**Mazza et al., 2022**).

Due to its chemical properties and redox-sensing ability, the catalytic cysteine in DJ-1 superfamily proteins holds critical importance during multi-stress responses (**Wilson, 2011**). In agreement with hDJ-1, C138A mutants of Hsp31 paralogs failed to repair macromolecules and detoxify RCS, suggesting a common multipurpose catalytic site.

## Yeast DJ-1 orthologs provide mitochondrial protection during carbonyl stress

MG and GO toxicity is one of the leading factors contributing to mitochondrial dysfunction in most neurodegenerative diseases (**Wang et al., 2019**). Mechanistically, they disrupt the mitochondrial morphology and electron transport chain and elevate ROS levels (**Videira and Castro-Caldas, 2018**). Although DJ-1 superfamily proteins display dynamic subcellular localization during the stress response (**Junn et al., 2009**; **Miller et al., 2003**; **Nair et al., 2018**), their physiological relevance in mitochondrial protection is not well-studied. In a previous report, we showed the maintenance of mitochondrial morphology by regulating the fusion-fission network by Hsp31 and Hsp34 (**Bankapalli et al., 2020**). Here, we provide evidence of mitochondrial maintenance by yeast DJ-1 orthologs, effectively through global regulation of glycation stress and attenuation of glycation damage on mitochondrial macromolecules. Interestingly, Hsp31 members redistribute into mitochondria in response to carbonyl toxicity that may substantially lower the glycation of mitochondrial proteome and DNA, contributing to enhanced mitochondrial survival. Moreover, the mitochondrial translocation of the paralogs could elevate the glycolate synthesis from glyoxal, a critical antioxidant that maintains mitochondrial membrane potential during stress conditions and re-establishes redox balance by producing GSH (**Diez et al., 2021**; **Toyoda et al., 2014**). Hence, under carbonyl stress, their absence leads to the accumulation of fragmented and dysfunctional mitochondria, significantly attenuating the availability of the ATP pool and contributing to detrimental mitochondrial health. The dysfunctional mitochondria are accompanied by poor maintenance of mitochondrial DNA quality, which may further suppress the activity of mitochondrial DNA-encoded respiratory complexes leading to compromised ATP generation. These pathogenic alterations of mitochondria are central to neurodegeneration and the rapid progression of PD.

Besides impaired carbonyl homeostasis, a major contributor to PD progression is α-synuclein toxicity (**Spillantini et al., 1997**). Its physiological role involves interaction with mitochondrial complex I and ATP synthase. However, MG glycated α-synuclein forms cytotoxic oligomers, which may induce mitophagy and mitochondrial fragmentation (**Faustini et al., 2017**; **Vicente Miranda et al., 2017**). Therefore, cells rely on DJ-1 member proteins that possess glyoxalase and deglycases to abrogate the toxic accumulation of glycated products. Our findings highlight a possible neuroprotective function of PD-associated hDJ-1 involving the repair of damaged macromolecules and mitochondrial quanlity control, promoting healthy maintenance of neuronal cells with intrinsically higher levels of RCS (**Angeloni et al., 2014**).

In summary, our findings significantly contribute to understanding the diversified response by Hsp31 paralogs in the field of carbonyl biology. Through the novel repair pathway identified in *S. cerevisiae*, the AGEs formation is efficiently suppressed, and mitochondrial performance is enhanced during chronic exposure to RCS. Although Hsp31 paralogs display stress-induced mitochondrial redistribution, their translocation pathway is poorly studied. Therefore, mechanistic insights into the

translocation process and their sub-compartment distribution would reveal critical mitochondrial-associated functions of Hsp31 paralogs. Also, further exploration of genotoxic damage induced in the absence of hsp31 paralogs may suggest how hDJ-1 encounters PD-linked chromatin aberrations.

# Materials and methods

## Strains and plasmid constructions

The yeast strains used in this study are mentioned in *Supplementary file 1*. The details of the primers and plasmids utilized in this study are described in *Supplementary file 1*. Knockouts of Hsp31 paralogs were generated through homologous recombination in haploid yeast strain BY4741 (*MATα his3Δ*1 *leu2Δ*0 *met15Δ*0 *ura3Δ*0). Deletion of *hsp32* (Δ32), *hsp33*(Δ33), and *glo1*(Δglo1), was performed by transforming suitable cassettes generated by primers P1-P4, constituting the *hphNT1* marker. Subsequently, the deletions were confirmed through PCR using primers P5-P8. The strains lacking hsp31 (Δ31) and hsp34 (Δ34) were previously generated in the lab. The double, triple (ΔT), and quadruple (ΔQ) deletion strains were generated by transforming DNA cassettes in Δ31Δ34. All the deletion strains were confirmed through PCR using primers P5-P8. RNR3 and Hsp31 paralogs expression was determined by genomically tagging at the C-terminus with heme agglutinin (HA) tag using primer P9-P11 and P12-P14, respectively, with *hphNT1* cassette as described above. Primers P15-P18 were used to tag RAD52 with GFP at the carboxy terminus in WT and ΔQ.

For overexpression studies, Hsp31 paralogs containing C-terminus heme agglutinin (HA) tag were cloned into pRS415 vector with GPD (glyceraldehyde-6-phosphate dehydrogenase) promotor and transformed into ΔQ. For protein expression and purification, Hsp31 family proteins were cloned into pRSF-Duet and purified with a previously published protocol (*Bankapalli et al., 2015*).

## Western blot analysis

Indicated respective strains grown till the mid-log phase ($A_{600}$~0.6) were treated with 10 mM MG or 15 mM GO for 12 hr in the culture tubes. Later, the cell lysates were prepared by incubating with 10% Trichloro acetic acid solution at 4 °C. Following acetone washes, 1 X SDS dye (50 mM Tris-HCl, pH 6.8, 2% SDS, 0.1% bromophenol blue, 10% glycerol, and 100 mM b-mercaptoethanol) and acid-washed glass bead was added and vortexed thoroughly. Subsequently, the samples were heated at 92 °C, and 30 μg of the sample was loaded. The lysates were resolved on 12% SDS-PAGE and transferred to the PVDF membrane. The membrane was blocked with 5% BSA in phosphate-buffered saline (PBST-Tween 0.05%). Later, the membrane was incubated overnight with anti-CML or anti-MAGE antibody 1:1000 at 4 °C. Following three washes with PBST, the membrane was incubated with the secondary antibody (1:10,000) at room temperature. Subsequently, the membrane was washed with PBST and exposed to luminol solutions (BIO-RAD). Equal loading of samples was confirmed through ponceau S staining of the membrane.

To determine the expression levels of Hsp31 paralogs in response to GO treatment, the WT strain containing genomic HA tag on the C-terminus of Hsp31 members was incubated in the absence or presence of 15 mM GO for 12 hr. Subsequently, the cell lysates were resolved on 15% SDS-PAGE and probed with an anti-HA antibody. Similarly, WT and ΔQ were incubated without or with 0.01% MMS, and 15 mM GO for 3 hr to estimate the expression of RNR3. The fold change in all experiments was calculated by measuring the band intensities using Multi Gauge V3.0 software.

## Nucleic acid glycation and DNA dot blot analysis

Yeast strains grown till the mid-exponential phase were treated with 10 mM MG or 15 mM GO in the culture tubes for 12 hr (*Brown, 2001*). Subsequently, genomic DNA was isolated using Wizard genomic DNA extraction kit (Promega, Cat No: A1120). For RNA isolation, cells were treated with 10 mM Tris pH 7.5, 1 mM EDTA, and 0.5% SDS (TES) buffer with hot acidic phenol for 1 hr at 65 °C. The resulting organic layer obtained after centrifugation was mixed with chloroform and centrifuged to acquire pure RNA. DNA and RNA were sheared through sonication and dotted (3 μg) on the nitrocellulose membrane (BIO-RAD, Cat No: 1620112). The membrane was baked at 80 °C for 2 hr and blocked-in phosphate-buffered saline (PBS-Tween 0.05%) with 5% BSA for 2 hr at room temperature. Consequently, the membrane was probed overnight with an anti-CML or anti-MAGE antibody. Following the washes, the membrane was incubated with a secondary anti-mouse antibody. The

membrane was exposed to luminol solutions. The membrane was stained with methylene blue to confirm equal loading of the samples.

## Polysome profiling

The experiment was performed using an established protocol (*Shaffer and Rollins, 2020*). Briefly, WT and ΔQ strains were grown till the mid-log phase and treated with 15 mM GO for 4 hr at 30 °C, and post-treatment cycloheximide was added to immobilize the ribosomes on mRNA. The cell lysates were prepared by adding lysis buffer (10 mM Tris pH 7.4, 100 mM NaCl, 30 mM MgCl$_2$, 100 µg/ml cycloheximide, 10 U RiboLock, and 1 mM PMSF) to the pellet and vortexed with glass beads. The lysates were loaded on SW41 tubes containing a 10% to 50% sucrose density gradient, followed by ultra-centrifugation. Later, the fractions were analyzed using a polysome profiler (Piston gradient fractionator; BIOCOMP). The area under the curve of polysome and monosome was calculated using Origin 8.0, and the ratio was plotted in Prism GraphPad 5.0.

## Phenotypic analysis and growth conditions

Individual yeast strains were grown till mid-log phase($A_{600}$~0.6) in YPD (yeast extract-1%, peptone-2%, and dextrose-2%) or synthetic dropout (SD) lacking leucine amino acid (0.67% yeast nitrogen base without amino acids, 0.072% Leu dropout supplement, and 2% dextrose). Consequently, cells were pelleted and serially diluted 10-fold until $10^{-5}$ dilution factor. Each dilution was spotted on the media plate without or with 15 mM GO. Parallelly, the cell pellet was treated with water or 10 mM MG for 6 hr, serially diluted, and spotted on media plates. The plates were incubated at 30 °C for 36 hr and imaged. Phenotype under DNA damaging agents was determined by spotting cells from the mid-exponential phase onto YPD media plates containing 0.01% MMS (methyl methanesulfonate) or 150 mM HU (hydroxyurea). Also, the cells were spotted on YPD media plates containing 15 mM GO or 0.01% MMS and plates with both genotoxins. Images of spot assay were taken at 36 hr. For growth assay on non-fermentable carbon source, mid-log phase cells were spotted on S.D. Glycerol (2%) media.

## *In vitro* glyoxalase and methylglyoxalase activity

A previously established protocol was used to determine the activity (*Bankapalli et al., 2015*; *Lee et al., 2016*). To determine the enzyme activity, 5 µg of purified Hsp31 paralogs WT and C138A mutants were incubated with 0.5 mM GO or MG in HEPES KOH pH 7 buffer for 30 min at 30 °C. Later, the reaction was terminated using 0.1% DNPH (2,4-dinitrophenylhydrazone), followed by adding 10% NaOH and incubating at 42 °C for 10 min. The results were calorimetrically analyzed using Biospectrometer (Eppendorf), 570 nm for GO, and 530 nm for MG. The kinetic parameters were determined by incubating Hsp31 members with varying GO concentrations, and the readings were noted at different time points. The experiments were performed in three biological replicates and triplicates. Subsequently, the values were plotted on Prism GraphPad 5.0 to estimate the $K_m$ and $V_{max}$.

## *In vitro* deglycation assay of DNA and Proteins

The deglycation of DNA and protein was performed using previously published protocols with minor modifications (*Richarme et al., 2017*). For DNA deglycation experiments, 500 µM dNTPs or 150 µM forward and reverse primers of yeast *sod1* (P19-P20) were incubated without or with GO/MG (2 mM) in HEPES KOH buffer pH 7 at 30 °C for 2 hr. Before supplementing the reactions with Hsp31 paralogs, the glycation mixtures were first diluted (6-fold) to minimize the free GO or MG levels. Post glycation reaction, 5 µg of purified Hsp31 paralogs and BSA were supplemented and incubated for 3 hr at 30 °C. The deglycase activity of Hsp31 paralogs was determined by subjecting the samples to a polymerase chain reaction (PCR) to amplify the *sod1* gene in the presence of Phusion enzyme high fidelity DNA polymerase.

To measure the protein deglycation activity, 2 µg of purified human Sod1 and Lysozyme were treated without or in the presence of 2 mM GO/MG in HEPES KOH buffer pH 7 at 30 °C for 2 hr (*Prasad et al., 2022*). Subsequently, the samples were diluted 6-fold to minimize GO or MG residual levels. Post glycation of protein samples, the reaction mixture was supplemented with 5 µg of purified Hsp31 paralogs and incubated for 3 hr for deglycation. The deglycation status of proteins was

determined by resolving the samples on 15% SDS-PAGE, followed by western analysis and immuno-detected with anti-CML or anti-MAGE antibodies.

## Genetic mutation and sequencing

Respective strains were grown till the mid-log phase and treated with 15 mM GO or 10 mM MG in the culture flasks for 24 hr at 30 °C. Total DNA was isolated as described above, and PCR amplified for *sod1*, *rad14*, and *cox2* ORFs using primers P19-P24. Subsequently, the amplicons were sequenced by the Sangar method, and the raw reads were aligned with the WT sequences. The number of mutations was determined using the multiple-sequence alignment tool Clustal Omega and the data was plotted on GraphPad Prism 5.0.

## Microscopic analysis

To analyse the formation of RAD52 GFP foci, WT and ΔQ cells were grown until the exponential phase and treated with 0.03% MMS and 15 mM GO for 1 hr. Subsequently, the $A_{600}$=0.5 OD cells were harvested and washed with 1 X PBS before spreading over 2% agarose pads. The number of foci were quantified and plotted.

The localization of yeast DJ-1 members was performed by genomically tagging with GFP at the carboxy terminus using primers P12-P14. Consequently, the cells were transformed with a pRS415$_{TEF}$ vector expressing MTS-mCherry to decorate mitochondria (*Bankapalli et al., 2015*). The transformed strains grown until the mid-exponential phase were harvested ($A_{600}$=0.5) and incubated with water or 15 mM GO for 2 hr. Parallelly, the cells were treated with water or 10 mM MG for 2 hr and harvested. The cell pellet was washed with 1 X PBS and processed for imaging. All the images were acquired in confocal microscopy (Olympus FV3000) with a 10 μm scale bar.

For mitochondrial DNA visualization, selected strains were grown till the mid-log phase and treated with 15 mM GO for 3 hr. Later, cells were harvested and stained with 10 μM SYTO 18 for 15 min at 30 °C, followed by visualization under confocal microscopy (Olympus FV3000). Images contain 10 μm scale bar.

## Analysis of mitochondrial mass

WT and ΔQ were grown to the early log phase and treated without or with 15 mM GO for 3 hr. Later, the cells were stained with 10 μm Nonyl Acridine Orange (NAO) for total mitochondrial mass or 10 μm TetraMethylRhodamine ethyl ester (TMRE) for measuring functional mass. The cells were subsequently washed and examined under a cytoFLEX flow cytometer.

## Measurement of mitochondrial ATP levels

A total of 30 μg of fractionated mitochondria from WT and ΔQ strains treated with GO were lysed and incubated with ATP detection solution from Mitochondrial ToxGlo Assay kit (Promega, Madison, USA). The luminescence was recorded using TECAN pro800.

## Chemicals and antibodies

Chemicals used in the study are Glyoxal (Sigma,128465), Methylglyoxal (Sigma, M0252), Methyl Methanesulfonate (TCI, M0369), Hydroxyurea (Sigma, H8627), Lysozyme (Sigma, 4403), and nitro-cellulose membrane (BIO-RAD,1620112). Antibodies used in the study are anti-CML (Carboxymethyl lysine; MAB3247 R&D Systems, Minneapolis, MN, USA), anti-MAGE (Argpyrimidine specific; ab243074 Abcam), anti-Tim23 (BD Bioscience), anti-HA antibody (GT4810, Sigma), secondary mouse (GE-bioscience).

## Miscellaneous

Mitochondria were isolated from cells treated with 15 mM GO or 10 mM MG for 16 hr using a previously published protocol (*Kumar et al., 2020*). For mitochondrial DNA, fractionated mitochondria were lysed and subjected to alkaline lysis DNA purification (*Sambrook and Russell, 2006*).

## Statistical analysis

Glycation intensities of DNA and proteins were measured through quantitation of complete dots or lanes, respectively, in Multi Gauge V3.0 software. The values were normalized with respect to WT in all

experiments and plotted in GraphPad Prism 5.0 software. Error bars represent the standard deviation derived from three biological replicates. One-way ANOVA with Dunnett's multiple comparisons test was used to determine significance analysis, comparing multiple columns against the WT or paired t-test for analysing single column. Asterisks used in the figures represent the following significance values: *, $p \leq 0.05$; **, $p \leq 0.01$; ***, $p \leq 0.001$; ****, $p \leq 0.0001$; NS, not significant.

## Acknowledgements

We thank Prof. Umesh Varshney (Department of Microbiology and Cell Biology, Indian Institute of Science, Bangalore, India) for the utilization of the polysome profile machine and Prof. Elizabeth A Craig (University of Wisconsin-Madison, USA) for yeast vectors (pRS415$_{TEF}$ and pRS415$_{GPD}$). We acknowledge the confocal microscope and flow cytometer facility (Department of Biochemistry, Indian Institute of Science, Bangalore, India). Prof. Patrick D'Silva acknowledges financial support from DST-SERB grant (CRG/2018/001988), Department of Biotechnology (DBT-IISC Partnership Program Phase-II, No. BT/PR27952/IN/22/212/2018), and DST-FIST Program-Phase III (No. SR/FST/LSII045/2016 G). Gautam Susarla and Priyanka Kataria acknowledge the fellowship from Indian Institute of Science-Bangalore, India. Amrita Kundu acknowledges the fellowship from the Department of Biotechnology-Research associate (DBT-RA), India.

---

## Additional information

### Funding

| Funder | Grant reference number | Author |
|---|---|---|
| Science and Engineering Research Board | CRG/2018/001988 | Patrick D'Silva |
| Department of Science and Technology, Ministry of Science and Technology, India | SR/FST/LSII045/2016-G | Patrick D'Silva |
| Department of Biotechnology, Ministry of Science and Technology, India | BT/PR27952/ IN/22/212/2018 | Patrick D'Silva |

The funders had no role in study design, data collection and interpretation, or the decision to submit the work for publication.

### Author contributions

Gautam Susarla, Conceptualization, Data curation, Formal analysis, Validation, Investigation, Visualization, Methodology, Writing - original draft, Writing – review and editing; Priyanka Kataria, Conceptualization, Data curation, Formal analysis, Visualization, Methodology, Writing – review and editing; Amrita Kundu, Data curation, Formal analysis, Validation, Methodology; Patrick D'Silva, Conceptualization, Resources, Formal analysis, Supervision, Funding acquisition, Validation, Investigation, Project administration, Writing – review and editing

### Author ORCIDs

Gautam Susarla http://orcid.org/0000-0002-3798-749X
Priyanka Kataria http://orcid.org/0000-0003-1860-4286
Amrita Kundu http://orcid.org/0000-0002-1499-7564
Patrick D'Silva http://orcid.org/0000-0002-1619-5311

### Decision letter and Author response

Decision letter https://doi.org/10.7554/eLife.88875.sa1
Author response https://doi.org/10.7554/eLife.88875.sa2

## Additional files

### Supplementary files
• Supplementary file 1. List of strains, primers, and plasmids used in this study.

• MDAR checklist

### Data availability
All data generated or analysed during this study are included in the manuscript and supporting file; Source Data files have been provided for all the Figures and Figure Supplements.

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
