## [Editor Report]

This paper reports the important discovery of a mechanism of cellular response to the reactive carbonyl species, which in yeast involves Hsp31, Hsp32, Hsp33, and Hsp34, all orthologs of DJ-1 protein, a determinant of Parkinson's disease. The evidence supporting the role of Hsp31-34 in preventing and repairing damage to mitochondria, proteins and nucleic acids by reactive carbonyl species is convincing. The study will be of interest to molecular cell biologists and beyond, as it helps to shed new light on the molecular basis of Parkinson's disease pathology.

---

## [Decision Letter]

**Decision letter after peer review:**

[Editors’ note: the authors submitted for reconsideration following the decision after peer review. What follows is the decision letter after the first round of review.]

Thank you for submitting the paper "*Saccharomyces cerevisiae* DJ-1 paralogs maintain genome integrity through glycation repair of nucleic acids and proteins" for consideration by *eLife*. Your article has been reviewed by 3 peer reviewers, and the evaluation has been overseen by a Reviewing Editor and a Senior Editor. The following individual involved in the review of your submission has agreed to reveal their identity: Carlo Vascotto (Reviewer #2).

Comments to the Authors:

We are sorry to say that, after consultation with the reviewers, we have decided that this work will not be considered further for publication by *eLife*.

The reviewers agree that in its current form the manuscript requires major revisions to strengthen the aspects that are new in this work, which will likely take considerably longer than a few months. The list of critical comments and suggestions to improve the manuscript is appended below. Given the high interest in the topic and the potential of the findings, we encourage you to consider *eLife* for the submission of a more mature version of the work.

*Reviewer #1 (Recommendations for the authors):*

In this study, Susarala and colleagues attempted to uncover the functional importance of 4 human DJ-1 paralogs in *Saccharomyces cerevisiae*. By utilizing yeast strains lacking different combinations of these paralogs and by re-introducing them individually via overexpression, they unequivocally answer the role of these DJ-1 homologs in protecting against different forms of glycation damage. Furthermore, their study offers mechanistic insight into how these paralogs both neutralize reactive carbonyl species and reverse glycation damage to DNA, RNA, and proteins. Finally, they show that these paralogs translocate to mitochondria during glycation stress, illustrating the important role of these homologs in protecting mitochondria from reactive carbonyl species.

Thus, this study firmly establishes the physiological role of the 4 yeast DJ-1 homologs Hsp31, Hsp32, Hsp33, and Hsp34. While these results have implications for how proteostatic stress is mitigated and proteostasis is maintained in *Saccharomyces cerevisiae*, they lack direct translatability to various disease models in which DJ-1 is suspected to play a physiological role. Moreover, the translocation of the paralogs to mitochondria during reactive carbonyl species stress is an interesting observation, but further studies are needed to fully elucidate the mechanistic importance of this process in different disease models.

1. For Figure 1A, given the translocation of the paralogs to mitochondria during glycation stress, it would be interesting to observe the growth of these strains on a non-fermentable carbon source.

2. For Figure 2J, why are the RNA glycation damage values for the various Hsp knockout strains significantly lower than Δ31 in Figure 2I? Given that these figures concern methylglyoxal-induced damage, shouldn't we expect the values to be more similar?

3. The in vitro activity assays are very well done, but it is the opinion of this reviewer that the results of the C138A mutants do not add significantly to the overall story. Perhaps these data are better suited for the supplement?

4. The confocal images in Figures 5I, 7, and 7-supplement 1 would benefit from the inclusion of rigorous quantification to show RAD52 foci formation and percent co-localization of the paralogs to mitochondria.

5. It is the opinion of the reviewer that the manuscript would benefit if Figure 7 —figure supplement 1 was somehow included in the main figure. Furthermore, it would be helpful if in the supplement it was shown that Hsp32, Hsp33, and Hsp34 do not translocate to mitochondria during methylglyoxal stress.

6. In the first paragraph of the discussion, you mention that Hsp33 has a different active site than Hsp31. Do you instead mean that Hsp32, Hsp33, and Hsp34 all have the same active site and that only Hsp31 has a different active site?

7. Lines 497-498 would benefit from the addition of citations.

8. On line 509, what do you mean by "stabilizes mitochondrial membrane potential"? Additional clarification and/or rephrasing would be helpful.

9. On lines 513-516, you highlight that the potential translocation of DJ-1 to mitochondria may help delay the progression of Parkinson's disease by attenuating glycation damage. However, this point warrants additional discussion and citations concerning how this may impact interactions of α-synuclein with mitochondria, and further how this may be important for the pathogenesis of Parkinson's disease.

10. On line 519, I would argue that the claim of conferring cytoprotection by preserving organellar integrity is unsubstantiated by the current study.

11. Lines 520-522 require citations.

12. It is the opinion of this reviewer that the Discussion section of this manuscript would benefit from the inclusion of a paragraph discussing both the limitations of the current study and potential avenues for follow-up studies.

*Reviewer #2 (Recommendations for the authors):*

With this study, Susarla et al. investigate a previously neglected family of yeast enzymes belonging to the DJ -1 superfamily of proteins. In particular, the authors focus on Hsp31 paralogs, which play a leading role in controlling the glycation of proteins and nucleic acids. The manuscript is logically organized and easy to read. The methodological approaches chosen are appropriate, and the data are presented logically and coherently, largely supporting the hypothesis.

The weaknesses that I noticed relate to:

1. The mitochondrial function of these proteins is not fully and adequately supported by the data presented.

2. Some controls necessary to support the experimental approach are missing.

3. The discrepancy between some images and the relative densitometric analysis.

In summary, the manuscript has a high degree of novelty, and the new findings greatly expand the current knowledge in the field.

Although the reported data are convincing, I think the authors should consider the following points to improve the quality of their work:

1. The impact of the results on mitochondrial maintenance is limited and not properly supported.

2. In Figure 7E, there are no markers showing that the analysis was performed on isolated mitochondria. The author should include at least one mitochondrial marker protein.

3. The sentence on page 21, line 505 "Interestingly, translocation of Hsp31 members into mitochondria substantially lowers the glycation of mitochondrial proteome and DNA,…" I cannot find any data showing a decrease in mtDNA glycation levels. The authors should measure the glycation level of isolated mtDNA under ΔQ and Hsp31 overexpression conditions.

4. Figure 2 G, H: I found no agreement between the image and relative densitometric analysis. e.g.: in panel G, the glycation level of DNA is reported to be 40-fold higher in Dglo1 than in WT. In panel I, an 8-fold increase is reported, but looking at the picture, it looks much more. The same is true for the other two panels (H and J). I also notice a discrepancy between the image and the bar graph in Figure 4C. The signal in samples 32 and 33 seems to me to be stronger than that in ΔQ. Since this is a representative image of biological replicates, it should also be representative of the data reported in the densitometric analysis. The authors should revise these data.

5. In Figure 3A-B, the authors use recombinant proteins to measure the enzymatic activities and kinetic parameters of Hsp31 paralogs. I suggest including a picture of an SDS-PAGE analysis of the recombinant proteins used.

*Reviewer #3 (Recommendations for the authors):*

The authors were trying to explore the mechanism of action of the DJ-1 paralogs Hsp31, Hsp32, Hsp33, and Hsp34 and whether there may be differences between these highly similar orthologs. They did a good job of showing that while for reactive carbonyl species, glyoxal, they behave similarly, but, for methylglyoxal, only Hsp31 was able to revert this damage.

My main criticism of the work is that for many of the other conclusions it is unclear how much this advances the field over previous work already published by this group and others. They have already previously published that Hsp31 has robust glyoxalase activity (Bankapalli et al. 2015). While others have shown that DJ-1 and its bacterial homologs can repair glycated nucleic acids and proteins (Richarme et al. 2017, Richarme et al. 2018). While an argument could be made that it is unclear if human DJ-1 and Hsp31 would have the same activity, they also previously showed that human DJ-1 can complement hsp31∆ (Bankapalli et al. 2015). They also mention how distribution to the mitochondria can alleviate glycation damage, but they already showed that DJ-1 orthologs redistribute to the mitochondria during stress (Bankapalli et al. 2015).

Another criticism is that some of their findings are overstated relative to the data they show here.

"…the glycation of RNA significantly reduced mRNA translation activity in the absence of yeast DJ-1 members". This is not directly shown. While they do see reduced mRNA translation activity while they see increased glycation of RNA, they have also shown that there are many levels of stress response induced by this treatment (DNA damage, mitochondrial stress). As a reduction of translation initiation is a common occurrence from a variety of stresses, it is unclear that it can be concluded that the reduced translational activity is due directly to the glycation of RNA and not a general stress response of the cell.

"DJ-1 orthologs provide robust organellar protection by redistributing into mitochondria to alleviate the glycation damage of mitochondrial DNA and proteins." "Here, we provide direct evidence of mitochondrial maintenance by yeast DJ-1 orthologs, effectively through robust redistribution under glycation stress." I would disagree that they show direct evidence that redistribution is necessary for the protective effects they see. While it is shown that these proteins relocalize to the mitochondria, it isn't shown that relocalization is required to protect the mitochondrial proteins. It could be that the glyoxalase-driven reduction of these reactive carbonyl species is sufficient to protect the mitochondria without translocation to the mitochondria.

To directly show that mitochondrial localization of Hsp31 is necessary to protect the mitochondria, the authors should repress relocalization to the mitochondria and test the effects on mitochondrial proteins. Do these proteins have a mitochondrial targeting sequence? What is known about how these proteins are targeted to the mitochondria?

[Editors’ note: further revisions were suggested prior to acceptance, as described below.]

Thank you for resubmitting your work entitled "*Saccharomyces cerevisiae* DJ-1 paralogs maintain genome integrity through glycation repair of nucleic acids and proteins" for further consideration by *eLife*. Your revised article has been evaluated by Detlef Weigel (Senior Editor) and a Reviewing Editor.

The manuscript has been improved but there are some remaining issues that need to be addressed, as outlined below:

Specifically, reviewer 1 requests further improvements as outlined below, which may be addressed by text changes or experimentally, upon the authors' decision, and described in the point-by-point response.

However, it is critical that the authors thoroughly address points no 4 and 5.

Altogether, as you will see from the appended comments, the reviewers are impressed by the largely improved version of the manuscript.

*Reviewer #1 (Recommendations for the authors):*

1. First, I thank the authors for their thorough and thoughtful replies to our original critiques. I feel they have done an excellent job fully addressing our concerns, and I believe the manuscript is greatly strengthened as a result.

2. In Figure 8B, NAO is used to stain the total mitochondrial mass +/- GO treatment. Is it known whether or not the loss of Hsp31-34 paralogs has any effect on cardiolipin levels, which may affect the measurement of mitochondrial mass? Did you try using MitoTracker, or another membrane potential-independent dye?

3. In Figure 8D, you measure mitochondrial ATP levels after GO treatment. Of course, AAC would be exporting ATP during the isolation procedure. Have you attempted to instead measure cellular ATP levels after GO treatment, perhaps in a media with a non-fermentable carbon source?

4. In some of the figures, the number of independent biological replicates is listed (i.e., n = 3). However, other figures, such as Figure 5 and Figure 5-supplement 1, it is missing. Furthermore, for Figure 5—figure supplement 1G, it is unclear if the quantification from multiple images (> 3) is from one or more independent biological replicates. The review would find it helpful if all figures list the number of independent biological replicates performed.

5. Lastly, for Figure 8—figure supplement 1, in the text you mention that mitochondrial health is compromised in δ Q, but δ T also shows impaired growth on a non-fermentable carbon source.

*Reviewer #2 (Recommendations for the authors):*

They have improved the manuscript by including data that loss of the Hsp31 paralogs induced reduced growth in respiratory conditions as well as that loss of the paralogs leads to mitochondrial dysfunction and fragmentation. The authors have also improved their manuscript by reducing some of the claims they were making.

*Reviewer #3 (Recommendations for the authors):*

The authors have adequately addressed all the issues/concerns raised by the three reviewers and have implemented the manuscript accordingly.

In my opinion, the topic is relevant and the data presented are of high quality, compelling, and overall add to the current knowledge of DJ-1 orthologs as detoxifying enzymes against lesions caused by reactive carbonyl species on nucleic acids and proteins. As is often the case, new information on molecular mechanisms raises further questions, such as the molecular mechanisms responsible for mitochondrial transport or the relevance to human studies, but these could be addressed in future studies.

---

## [Author Response]

[Editors’ note: the authors resubmitted a revised version of the paper for consideration. What follows is the authors’ response to the first round of review.]

Comments to the Authors:Reviewer #1 (Recommendations for the authors):In this study, Susarala and colleagues attempted to uncover the functional importance of 4 human DJ-1 paralogs in *Saccharomyces cerevisiae*. By utilizing yeast strains lacking different combinations of these paralogs and by re-introducing them individually via overexpression, they unequivocally answer the role of these DJ-1 homologs in protecting against different forms of glycation damage. Furthermore, their study offers mechanistic insight into how these paralogs both neutralize reactive carbonyl species and reverse glycation damage to DNA, RNA, and proteins. Finally, they show that these paralogs translocate to mitochondria during glycation stress, illustrating the important role of these homologs in protecting mitochondria from reactive carbonyl species.

We thank the reviewer for summarising the critical highlights of our study.

Thus, this study firmly establishes the physiological role of the 4 yeast DJ-1 homologs Hsp31, Hsp32, Hsp33, and Hsp34. While these results have implications for how proteostatic stress is mitigated and proteostasis is maintained in *Saccharomyces cerevisiae*, they lack direct translatability to various disease models in which DJ-1 is suspected to play a physiological role. Moreover, the translocation of the paralogs to mitochondria during reactive carbonyl species stress is an interesting observation, but further studies are needed to fully elucidate the mechanistic importance of this process in different disease models.

Thank you for your response. We agree that the emerging functions of Hsp31 paralogs are fascinating and need further examination in human DJ-1 with desired model organisms to reveal its physiological significance. However, our study provides a possible mechanism of how human DJ-1 contributes to cellular protection during carbonyl toxicity. Indeed, the translocation of Hsp31 paralogs into mitochondria is interesting. Further experiments need to be performed to better understand the diverse response of DJ-1 and its mechanism to support mitochondrial health.

1. For Figure 1A, given the translocation of the paralogs to mitochondria during glycation stress, it would be interesting to observe the growth of these strains on a non-fermentable carbon source.

Thank you for your suggestion. We have performed the phenotypic growth analysis of all the strains from Figure 1A on a non-fermentable carbon source (glycerol), comparing it with dextrose (fermentable carbon source) media. This result was incorporated into the revised manuscript. Refer to Figure 8figure Supplement 1.

2. For Figure 2J, why are the RNA glycation damage values for the various Hsp knockout strains significantly lower than Δ31 in Figure 2I? Given that these figures concern methylglyoxal-induced damage, shouldn't we expect the values to be more similar?

We apologise for this mistake. The blot from figure 2I displayed a saturated signal, making it difficult to analyze the result. In the revised manuscript, it has been replaced with a blot having appropriate exposure, and the RNA glycation levels from Δ31 of figure 2I are almost similar to deletion strains of figure 2J, which is also supported by the densitometric graph.

3. The in vitro activity assays are very well done, but it is the opinion of this reviewer that the results of the C138A mutants do not add significantly to the overall story. Perhaps these data are better suited for the supplement?

We thank the reviewer for the valuable input. The macromolecular deglycation by the C138A mutant of Hsp31 and its paralogs are now moved to supplement data. Refer to Figure 5—figure supplement A-D and Figure 6—figure supplement A-D.

4. The confocal images in Figures 5I, 7, and 7-supplement 1 would benefit from the inclusion of rigorous quantification to show RAD52 foci formation and percent co-localization of the paralogs to mitochondria.

Thank you for the technical suggestion. We have quantitated the RAD52 foci formation from Figure 5I, and the graph is provided in the revised manuscript in Figure 5—figure supplement G. For percentage localization of Hsp31 paralogs into mitochondria from Figure 7 (A to D) and Figure 7-supplement 1, we performed westerns analysis without and with glyoxal and methylglyoxal treatments. Subsequently, the mitochondria were isolated and examined for the fold change in the localization of Hsp31 paralogs. These results are incorporated in the revised manuscript in Figure 7—figure supplement 1A and B.

5. It is the opinion of the reviewer that the manuscript would benefit if Figure 7 —figure supplement 1 was somehow included in the main figure. Furthermore, it would be helpful if in the supplement it was shown that Hsp32, Hsp33, and Hsp34 do not translocate to mitochondria during methylglyoxal stress.

We thank the reviewer for the constructive advice. Figure 7—figure supplement 1 is now included in Figure 7 E of the main figures. We performed microscopic visualization studies to show that Hsp32, Hsp33, and Hsp34 do not enrich in mitochondria during methylglyoxal stress. The results are presented in the revised manuscript. Refer to Figure 7—figure supplement 1 C to E.

6. In the first paragraph of the discussion, you mention that Hsp33 has a different active site than Hsp31. Do you instead mean that Hsp32, Hsp33, and Hsp34 all have the same active site and that only Hsp31 has a different active site?

We apologize for creating confusion in the statement. The line states that Hsp32, Hsp33, and Hsp34 have a similar active site, which differs from Hsp31. In the revised manuscript, we have changed the statement to the following “Since Hsp32 and Hsp34 are structurally uncharacterized, we considered overlapping structural features with Hsp33 for comparative analysis due to its solved crystal structure and close sequence similarity. Strikingly, Hsp32, Hsp33, and Hsp34 possess a similar active site that offers different substrate specificity than Hsp31.”

7. Lines 497-498 would benefit from the addition of citations.

Thank you for your suggestions. We have added a relevant citation at the end of the statement.

8. On line 509, what do you mean by "stabilizes mitochondrial membrane potential"? Additional clarification and/or rephrasing would be helpful.

We apologize for the inconvenience. The statement has been rephrased to the following in the revised manuscript “Moreover, such redistribution of the paralogs could elevate the glycolate synthesis from glyoxal, a critical antioxidant that maintains mitochondrial membrane potential during stress conditions and re-establishes redox balance by producing GSH”.

9. On lines 513-516, you highlight that the potential translocation of DJ-1 to mitochondria may help delay the progression of Parkinson's disease by attenuating glycation damage. However, this point warrants additional discussion and citations concerning how this may impact interactions of α-synuclein with mitochondria, and further how this may be important for the pathogenesis of Parkinson's disease.

Thank you for your critical insight. In the revised manuscript, we have discussed the interaction of α-synuclein with mitochondria and the potential effects of glycated α-synuclein in the pathological progression of Parkinson’s disease. Refer to line number 567-571.

10. On line 519, I would argue that the claim of conferring cytoprotection by preserving organellar integrity is unsubstantiated by the current study.

We apologize for the mistake. The word cytoprotection has been removed in the current version of the manuscript.

11. Lines 520-522 require citations.

We apologise for the inconvenience. We have provided appropriate citations at the end of the statement in the revised manuscript.

12. It is the opinion of this reviewer that the Discussion section of this manuscript would benefit from the inclusion of a paragraph discussing both the limitations of the current study and potential avenues for follow-up studies.

Thank you for your valuable suggestion. We have now included a paragraph at the end of the discussion section explaining the limitations of the current study and the need to discover new findings to address potential and emerging avenues in the field.

Reviewer #2 (Recommendations for the authors):With this study, Susarla et al. investigate a previously neglected family of yeast enzymes belonging to the DJ -1 superfamily of proteins. In particular, the authors focus on Hsp31 paralogs, which play a leading role in controlling the glycation of proteins and nucleic acids. The manuscript is logically organized and easy to read. The methodological approaches chosen are appropriate, and the data are presented logically and coherently, largely supporting the hypothesis.The weaknesses that I noticed relate to:1. The mitochondrial function of these proteins is not fully and adequately supported by the data presented.2. Some controls necessary to support the experimental approach are missing.3. The discrepancy between some images and the relative densitometric analysis.

As per the reviewer’s suggestions, we have performed multiple sets of experiments to strengthen the role of Hsp31 paralogs in mitochondrial protection and maintenance, which has been elaborately described in response to comment 1 below. Also, the discrepancies in the images and the relative densitometric analysis have been corrected and incorporated in the revised manuscript.

In summary, the manuscript has a high degree of novelty, and the new findings greatly expand the current knowledge in the field.

Thank you for understanding the potential of our work and its additional contribution to the existing knowledge.

Although the reported data are convincing, I think the authors should consider the following points to improve the quality of their work:1. The impact of the results on mitochondrial maintenance is limited and not properly supported.

We thank the reviewer for their critical analysis and suggestions. We have now included a large data set (refer to Figure 8) related to mitochondrial maintenance by Hsp31 paralogs. A comprehensive study investigating various parameters of mitochondria, as listed below, was performed:

Phenotypic analysis on non-fermentable carbon source (glycerol) to decipher the involvement of Hsp31 paralogs in mitochondrial maintenance [Figure 8—figure supplement 1]Microscopic studies of mitochondrial morphologies to visualize altered mitochondrial network in the presence of glycation stress [Figure 8 A]Quantitative measurement of total and functional mitochondrial mass to investigate the effect of glycation toxicity on global mitochondrial content [Figure 8 B and C].Measurement of mitochondrial ATP levels to assess the role of Hsp31 paralogs in preserving mitochondrial integrity [Figure 8 D].Microscopic visualization of mitochondrial DNA to study the impact of glyoxal glycation on DNA maintenance in the absence of DJ-1 members [Figure 8 E].

2. In Figure 7E, there are no markers showing that the analysis was performed on isolated mitochondria. The author should include at least one mitochondrial marker protein.

Thank you for the insights. We have now included the western analysis of mitochondria used in Figure 7E to show its purity and quality. The mitochondria isolates were probed with Tim23 as mitochondrial control and Ydj-1 for cytosol control. Refer to Figure 7—figure supplement 1F.

3. The sentence on page 21, line 505 "Interestingly, translocation of Hsp31 members into mitochondria substantially lowers the glycation of mitochondrial proteome and DNA,…" I cannot find any data showing a decrease in mtDNA glycation levels. The authors should measure the glycation level of isolated mtDNA under ΔQ and Hsp31 overexpression conditions.

We thank the reviewer for the valuable suggestions. The glycation levels of mtDNA under ΔQ and Hsp31 overexpression conditions were measured from purified mitochondria, and the data (refer to Figure 7G) has been included in the revised manuscript.

4. Figure 2 G, H: I found no agreement between the image and relative densitometric analysis. e.g.: in panel G, the glycation level of DNA is reported to be 40-fold higher in Dglo1 than in WT. In panel I, an 8-fold increase is reported, but looking at the picture, it looks much more. The same is true for the other two panels (H and J). I also notice a discrepancy between the image and the bar graph in Figure 4C. The signal in samples 32 and 33 seems to me to be stronger than that in ΔQ. Since this is a representative image of biological replicates, it should also be representative of the data reported in the densitometric analysis. The authors should revise these data.

We apologize for the inconvenience due to densitometric discrepancies. In the current manuscript, we have re-quantitated the western blots and revised all the densitometric graphs, whose values are now comparable to representative images. Figure 4C is replaced with better quality blot and exposure, where the glycation levels of Δ*Q* is comparable to overexpression strains of 32, 33, and 34.

5. In Figure 3A-B, the authors use recombinant proteins to measure the enzymatic activities and kinetic parameters of Hsp31 paralogs. I suggest including a picture of an SDS-PAGE analysis of the recombinant proteins used.

Thank you for your suggestion. We have now included the image of the SDS-PAGE analysis of purified Hsp31 paralog proteins. Refer to Figure 3—figure supplement 1A.

Reviewer #3 (Recommendations for the authors):The authors were trying to explore the mechanism of action of the DJ-1 paralogs Hsp31, Hsp32, Hsp33, and Hsp34 and whether there may be differences between these highly similar orthologs. They did a good job of showing that while for reactive carbonyl species, glyoxal, they behave similarly, but, for methylglyoxal, only Hsp31 was able to revert this damage.

We thank the reviewer for appreciating the work and describing the key highlights of our study.

My main criticism of the work is that for many of the other conclusions it is unclear how much this advances the field over previous work already published by this group and others. They have already previously published that Hsp31 has robust glyoxalase activity (Bankapalli et al. 2015).

We thank the reviewer for the critical analysis. The key highlights of the present study show are listed below for the first time.

We examined the enzymatic properties of the unexplored members of Hsp31 minifamily proteins, where Hsp32, Hsp33, and Hsp34 possess glyoxalase activity. At the same time, Hsp31 has glyoxalase and methylglyoxalase activity, suggesting that multiple paralogs of DJ-1 could aid enhanced protection.We show that the absence of yeast DJ-1 members induces severe glycation of DNA, RNA, and protein, along with reduced mRNA translation activity under carbonyl stress.Our studies indicate that DJ-1 members are involved in genome maintenance as their disruption upregulates DNA damage response and enhances mutation frequency.Our report provides evidence of glycation repair pathways in *S. cerevisiae*. Furthermore, the paralogs have substrate preference enabling Hsp31 to deglycate methylglyoxal and glyoxal glycated macromolecules, while Hsp32, Hsp33, and Hsp34 are specific to glyoxal glycated substrates.For the first time, we show that yeast DJ-1 members translocate into mitochondria during carbonyl stress. They also influence global mitochondrial content and ATP levels during glycation toxicity.

The reviewer refers to (Bankapalli et al.) where authors discovered methylglyoxalase activity of Hsp31 alone and its role in oxidative stress. In the current report, we have investigated more comprehensively the physiological relevance of the unexplored members (Hsp32, Hsp33, and Hsp34) of Hsp31 paralogs in protection against glycation stress essentially through enzymatic functions.

While others have shown that DJ-1 and its bacterial homologs can repair glycated nucleic acids and proteins (Richarme et al. 2017, Richarme et al. 2018). While an argument could be made that it is unclear if human DJ-1 and Hsp31 would have the same activity, they also previously showed that human DJ-1 can complement hsp31∆ (Bankapalli et al. 2015).

We thank the reviewer for the valuable response. Although human DJ-1 and Hsp31 share partial functional conservation under oxidative stress, as reported by Bankapalli et al. 2015, very few studies were performed to understand the role of human DJ-1 under carbonyl stress and its implications in mitochondrial functioning. The absence of Hsp31 paralogs collectively induces glycation damage of macromolecules and promotes mitochondrial impairments through altered mitochondrial morphology, loss of functional mitochondria, and depleted ATP levels. Refer to Figure 8.

In summary, our report gives insights into how hDJ-1 could govern cellular and organellar health during carbonyl toxicity, a leading factor for the progression of Parkinson’s disease. Human DJ-1 possibly contributes through reverting glycation damage of DNA, RNA, and proteins. Further, it might translocate into mitochondria under carbonyl stress to abrogate glyoxal toxicity and maintain functional mitochondria.

They also mention how distribution to the mitochondria can alleviate glycation damage, but they already showed that DJ-1 orthologs redistribute to the mitochondria during stress (Bankapalli et al. 2015).

Thank you for the response. The authors from Bankapalli et al. 2015 investigated the redistribution of Hsp31 alone into mitochondria during oxidative stress. On the other hand, our report highlights the mitochondrial translocation of all the Hsp31 members in response to glycation toxicity, together alleviating the glycation damage.

Another criticism is that some of their findings are overstated relative to the data they show here."the glycation of RNA significantly reduced mRNA translation activity in the absence of yeast DJ-1 members". This is not directly shown. While they do see reduced mRNA translation activity while they see increased glycation of RNA, they have also shown that there are many levels of stress response induced by this treatment (DNA damage, mitochondrial stress). As a reduction of translation initiation is a common occurrence from a variety of stresses, it is unclear that it can be concluded that the reduced translational activity is due directly to the glycation of RNA and not a general stress response of the cell.

We apologize for the mistake. Since there was enhanced RNA glycation, we investigated its effects on mRNA translation efficiency and observed a reduction. However, as the reviewer suggested, the attenuation in translation could be a general stress response. Therefore, we have rephrased the statement to the following “Moreover, in response to glycation stress, mRNA translation activity was significantly reduced in the absence of yeast DJ-1 members”.

"DJ-1 orthologs provide robust organellar protection by redistributing into mitochondria to alleviate the glycation damage of mitochondrial DNA and proteins." "Here, we provide direct evidence of mitochondrial maintenance by yeast DJ-1 orthologs, effectively through robust redistribution under glycation stress." I would disagree that they show direct evidence that redistribution is necessary for the protective effects they see. While it is shown that these proteins relocalize to the mitochondria, it isn't shown that relocalization is required to protect the mitochondrial proteins. It could be that the glyoxalase-driven reduction of these reactive carbonyl species is sufficient to protect the mitochondria without translocation to the mitochondria.

We thank the reviewer for the critical analysis of our work. We observed a 2-3-fold increase in mitochondrial translocation of Hsp31 paralogs during carbonyl stress, which may abrogate macromolecular glycation (Figure 7—figure supplement 1A, E). However, as the reviewer pointed out, we need to perform extensive studies in the future to understand the localization mechanism of Hsp31 paralogs and their relevance in providing mitochondrial protection. Moreover, we have rephrased the sentence in the revised manuscript. Refer to lines 551-553.

To directly show that mitochondrial localization of Hsp31 is necessary to protect the mitochondria, the authors should repress relocalization to the mitochondria and test the effects on mitochondrial proteins. Do these proteins have a mitochondrial targeting sequence? What is known about how these proteins are targeted to the mitochondria?

Thank you for your critical question and suggestion. This has been a debate over the past 2 decades on how DJ-1 translocates into the mitochondrial compartment without having a welldefined N-terminal targeting sequence (Kojima et al.; Maita et al.). However, it is not surprising that many proteins show a dual localization under specific circumstances (such as oxidative stress, glyoxal stress), presumably by forming a complex with the resident mitochondrial proteins or due to posttranslational modifications. We believe that Hsp31 paralogs from yeast exhibit a similar phenotype (Elliott and Volkert; Heo et al.).

By considering reviewer suggestions constructively, we made several attempts to address this question past 6 months. The following are salient features of our findings:

Considering the existing literature and information available in online databases, strikingly, there is no evidence of MTS in yeast Hsp31 paralogs, and their translocation mechanism needs to be better studied. On the other hand, mutations in the N-terminal region of hDJ-1 affect mitochondrial translocation (Maita et al.).

Based on sequence alignment, we attempted to make several mutants (in combination) on the N-terminal region of Hsp31 and truncated N-terminal 12 residues. Subsequently, they were tested for phenotype and mitochondrial redistribution by probing isolated mitochondria from cells treated with methylglyoxal. Upon Western analysis, we observed an altered localization pattern in the Hsp31 mutants, comparable to WT Hsp31, but no repression was observed.

Moreover, the mutants showed enhanced localization of the protein in the absence of MG stress which is intriguing See Author response image 1, right K4A and K5A (lane 2 and 3) mutants compared to WT (lane 1). This suggests that the N-terminal residues play a role in redistributing the Hsp31 into the mitochondria. Moreover, we attempted to delete the protein's N terminal portion (truncated) to test the hypothesis. However, we could not detect the N-terminal truncated Hsp31 through the Western analysis.

However, elucidating the exact mechanism of how the N-terminal sequence regulates mitochondrial entry will be a larger question that requires a large time frame in the future.

**Author response image 1. sa2fig1:** 

ReferencesBankapalli, K. et al. "Robust Glyoxalase Activity of Hsp31, a Thij/Dj-1/Pfpi Family Member Protein, Is Critical for Oxidative Stress Resistance in *Saccharomyces cerevisiae*." *J*

*Biol Chem*, vol. 290, no. 44, 2015, pp. 26491-26507, doi:10.1074/jbc.M115.673624.

Elliott, N. A. and M. R. Volkert. "Stress Induction and Mitochondrial Localization of Oxr1 Proteins in Yeast and Humans." *Mol Cell Biol*, vol. 24, no. 8, 2004, pp. 3180-3187, doi:10.1128/mcb.24.8.3180-3187.2004.

Heo, J. M. et al. "A Stress-Responsive System for Mitochondrial Protein Degradation." *Mol Cell*, vol. 40, no. 3, 2010, pp. 465-480, doi:10.1016/j.molcel.2010.10.021.

Kojima, W. et al. "Unexpected Mitochondrial Matrix Localization of Parkinson's DiseaseRelated Dj-1 Mutants but Not Wild-Type Dj-1." *Genes Cells*, vol. 21, no. 7, 2016, pp. 772-788, doi:10.1111/gtc.12382.

Maita, C. et al. "Monomer Dj-1 and Its N-Terminal Sequence Are Necessary for Mitochondrial Localization of Dj-1 Mutants." *PLoS One*, vol. 8, no. 1, 2013, p. e54087, doi:10.1371/journal.pone.0054087.

[Editors’ note: what follows is the authors’ response to the second round of review.]

The manuscript has been improved but there are some remaining issues that need to be addressed, as outlined below:Specifically, reviewer 1 requests further improvements as outlined below, which may be addressed by text changes or experimentally, upon the authors' decision, and described in the point-by-point response.However, it is critical that the authors thoroughly address points no 4 and 5.Altogether, as you will see from the appended comments, the reviewers are impressed by the largely improved version of the manuscript.Reviewer #1 (Recommendations for the authors):1. First, I thank the authors for their thorough and thoughtful replies to our original critiques. I feel they have done an excellent job fully addressing our concerns, and I believe the manuscript is greatly strengthened as a result.

We are grateful for such a response. We thank the reviewer for critical comments and insightful suggestions, which helped in reorganizing and strengthening various aspects of the manuscript.

2. In Figure 8B, NAO is used to stain the total mitochondrial mass +/- GO treatment. Is it known whether or not the loss of Hsp31-34 paralogs has any effect on cardiolipin levels, which may affect the measurement of mitochondrial mass? Did you try using MitoTracker, or another membrane potential-independent dye?

Thank you for your comment. NAO dye was used to determine the total mitochondrial mass in Figure 8B, which stains the cardiolipin of mitochondria. However, there are no studies performed to estimate the cardiolipin levels in the absence of Hsp31 paralogs and hDJ-1. Besides, our microscopic data from Figure 8A, where cells are episomally expressing mCherry in a plasmid, also indicates that loss of Hsp31 paralogs alters mitochondrial content.

As suggested by the reviewer, we have utilized MitoTracker deep red, a potential dependent stain that is independent of cardiolipin levels, to measure mitochondrial mass (Xiao et al.). Since it is potential dependent, we observed a similar result comparable to TMRE staining (see Author response image 2, compare with Figure 8C), with ∆*Q* showing a higher fold of mitochondrial dysfunction than WT in the presence of GO treatment.

**Author response image 2. sa2fig2:** Estimating mitochondrial mass through MitoTracker deep red. WT and ∆*Q* cells from the mid-log phase were incubated without or with GO, and cells were stained with MitoTracker deep red, followed by flow cytometric analysis.

3. In Figure 8D, you measure mitochondrial ATP levels after GO treatment. Of course, AAC would be exporting ATP during the isolation procedure. Have you attempted to instead measure cellular ATP levels after GO treatment, perhaps in a media with a non-fermentable carbon source?

Thank you for the suggestion. We did not measure the cellular ATP levels since the global ATP pool is contributed through multiple biochemical pathways besides mitochondrial respiration (Dunn and Grider). Moreover, it is difficult to conclude glycation-induced mitochondrial damage from an altered total cellular ATP pool. Therefore, we isolated mitochondria devoid of cytosolic fraction and estimated ATP levels which directly correlate to mitochondrial respiration and health.

4. In some of the figures, the number of independent biological replicates is listed (i.e., n = 3). However, other figures, such as Figure 5 and Figure 5-supplement 1, it is missing. Furthermore, for Figure 5—figure supplement 1G, it is unclear if the quantification from multiple images (> 3) is from one or more independent biological replicates. The review would find it helpful if all figures list the number of independent biological replicates performed.

We apologize for the inconvenience. In the revised manuscript, we have included the respective number of independent biological replicates in their corresponding figure legends.

5. Lastly, for Figure 8—figure supplement 1, in the text you mention that mitochondrial health is compromised in δ Q, but δ T also shows impaired growth on a non-fermentable carbon source.

We apologize for our mistake. We have incorporated the following statement in the revised manuscript “Although we observe no growth difference in dextrose media, the collective absence of Hsp31 paralogs in ΔT and ΔQ induced sensitivity to glycerol media”.

Reviewer #2 (Recommendations for the authors):They have improved the manuscript by including data that loss of the Hsp31 paralogs induced reduced growth in respiratory conditions as well as that loss of the paralogs leads to mitochondrial dysfunction and fragmentation. The authors have also improved their manuscript by reducing some of the claims they were making.

We thank the reviewer for appreciating our work and giving us the opportunity to corroborate the role of hsp31 paralogs in mitochondrial maintenance. These recommendations further strengthened our findings and contributions to the existing knowledge of DJ-1 family proteins.

Reviewer #3 (Recommendations for the authors):The authors have adequately addressed all the issues/concerns raised by the three reviewers and have implemented the manuscript accordingly.In my opinion, the topic is relevant and the data presented are of high quality, compelling, and overall add to the current knowledge of DJ-1 orthologs as detoxifying enzymes against lesions caused by reactive carbonyl species on nucleic acids and proteins. As is often the case, new information on molecular mechanisms raises further questions, such as the molecular mechanisms responsible for mitochondrial transport or the relevance to human studies, but these could be addressed in future studies.

We greatly appreciate the reviewer’s thoughtful insight into the mechanism of mitochondrial translocation of Hsp31 paralogs. We agree with the reviewer that the translocation mechanism is indeed interesting, and the molecular mechanisms could be perused in future studies involving disease models.

References

Dunn, J. and M. H. Grider. "Physiology, Adenosine Triphosphate." *Statpearls*, StatPearls Publishing Copyright 2023, StatPearls Publishing LLC., 2023.

Xiao, B. et al. "Flow Cytometry-Based Assessment of Mitophagy Using Mitotracker." *Front Cell Neurosci*, vol. 10, 2016, p. 76, doi:10.3389/fncel.2016.00076.